# Is image-to-image translation the panacea for multimodal image registration? A comparative study

Jiahao Lu [1,2], Johan Öfverstedt [1], Joakim Lindblad[1] *, Nataša Sladoje [1]

**1** MIDA Group, Department of Information Technology, Uppsala University, Uppsala, Sweden, **2** IMAGE Section, Department of Computer Science, University of Copenhagen, Copenhagen, Denmark

* joakim.lindblad@it.uu.se

**Data Availability Statement:** All used datasets are openly shared on Zenodo: https://zenodo.org/record/5557568 (DOI: 10.5281/zenodo.5557568).

## Abstract

Despite current advancement in the field of biomedical image processing, propelled by the deep learning revolution, multimodal image registration, due to its several challenges, is still often performed manually by specialists. The recent success of image-to-image (I2I) translation in computer vision applications and its growing use in biomedical areas provide a tempting possibility of transforming the multimodal registration problem into a, potentially easier, monomodal one. We conduct an empirical study of the applicability of modern I2I translation methods for the task of rigid registration of multimodal biomedical and medical 2D and 3D images. We compare the performance of four Generative Adversarial Network (GAN)-based I2I translation methods and one contrastive representation learning method, subsequently combined with two representative monomodal registration methods, to judge the effectiveness of modality translation for multimodal image registration. We evaluate these method combinations on four publicly available multimodal (2D and 3D) datasets and compare with the performance of registration achieved by several well-known approaches acting directly on multimodal image data. Our results suggest that, although I2I translation may be helpful when the modalities to register are clearly correlated, registration of modalities which express distinctly different properties of the sample are not well handled by the I2I translation approach. The evaluated representation learning method, which aims to find abstract image-like representations of the information shared between the modalities, manages better, and so does the Mutual Information maximisation approach, acting directly on the original multimodal images. We share our complete experimental setup as open-source (https://github.com/MIDA-group/MultiRegEval), including method implementations, evaluation code, and all datasets, for further reproducing and benchmarking.

## 1 Introduction

We are witnessing a growing popularity of approaches which combine information from different imaging modalities [1] to maximise extracted information about an object of interest, particularly within life sciences. Different imaging techniques offer complementary

**Funding:** The authors are financially supported by the Wallenberg AI, Autonomous Systems and Software Program (WASP) AI-Math initiative (NS, JL), Vinnova, Sweden's Innovation Agency (projects 2017-02447 and 2020-03611) (JL, NS, JO), and the Swedish Research Council (project 2017-04385) (JL). This article was facilitated by the COST Action COMULIS (CA17121), supported by COST (European Cooperation in Science and Technology) (NS, NL). The funders had no role in study design, data collection and analysis, decision to publish, or preparation of the manuscript.

**Competing interests:** The authors have declared that no competing interests exist.

information about structure, function, dynamics, and molecular composition of a sample. To efficiently utilise and fuse such heterogeneous information, acquired images have to be spatially aligned, a process known as image registration.

Numerous methods have been proposed for automatic monomodal (intramodal) registration [2–4], typically based on comparison of intensity patterns in images (intensity-based approaches), or on finding correspondence between characteristic details in the images (feature-based methods). However, multimodal (intermodal) image registration is much more challenging. In cases when image appearances differ significantly between modalities, e.g., due to different underlying physical principles of imaging, assessing image similarity and finding correspondences across modalities are, in general, very difficult tasks.

Multimodal registration is often required and performed in medical and satellite imagery, where valuable information is in many situations acquired using several different sensors [5]. Due to the inherent discrepancy in intensity values in images acquired by different modalities, it is difficult to define generally applicable methods. Existing approaches are, therefore, often specific to a particular application (e.g., relying on anatomical properties of specific organs), and typically restricted to only a few combinations of modalities. Applications in biomedical image analysis bring in further challenges: (i) there exists a much greater variety of modalities (e.g., tens of different types of microscopies), (ii) specimens exhibit much wider variety, (iii) imaged objects (e.g., tissue) often lack distinctive structures (as opposed to, e.g., organs) which could be used for establishing correspondences, (iv) acquired data often consist of very large images (easily reaching TB-size), necessitating fast methods with low memory requirements and also limiting the feasibility of high-quality manual annotation. Direct application of conventional methods is most often not sufficient, even in relatively simple scenarios. As a consequence, biomedical image registration is nowadays still often performed manually by specialists [6].

Deep learning-based methods have, since their renaissance, revolutionised the field of computer vision. Significant results with this type of methods have also been reached in the field of medical image registration, as summarised in recent surveys [6–9]. The main considered approaches include learning of similarity measures, direct prediction of the transformation, and image-to-image (I2I) translation approaches. Methods based on direct prediction of the transformation have so far mostly provided an improvement of the run-time, while exhibiting registration performance that is (at best) on par with traditional methods [10, 11]. A related, but different recent method for multimodal rigid registration [12] uses a large neural network as a feature extractor, training a network to recover a transformation from a single image. This method, however, fails in case a canonical space does not exist, which holds for a majority of the imaging scenarios considered in this study. Generative Adversarial Network (GAN)-based methods have shown impressive performance in I2I translation tasks, primarily on natural images, [13–17]. They are offering the possibility of converting multimodal registration into a, presumably less challenging, monomodal problem which can be addressed by a wide range of monomodal registration methods; this idea is illustrated in Fig 1. This concept has been shown promising in many multimodal registration tasks, even before the GAN-based methods became a standard tool [18–20]. GAN-based approaches are receiving increasing attention in medical image registration [7, 8, 21], and the progression to bioimage analysis seems to be a natural step. A recent survey [22] overviews several usage scenarios of GANs in the field of biomedical image analysis, primarily targeting data generation aiming to overcome challenges of annotation in digital pathology. However, quantitative results related to image registration, and particularly, in multimodal settings on multiple datasets, are still lacking. The potential of GANs in these scenarios is still to be explored, as pointed out in a recent overview of the application of deep learning in bioimage analysis [23].

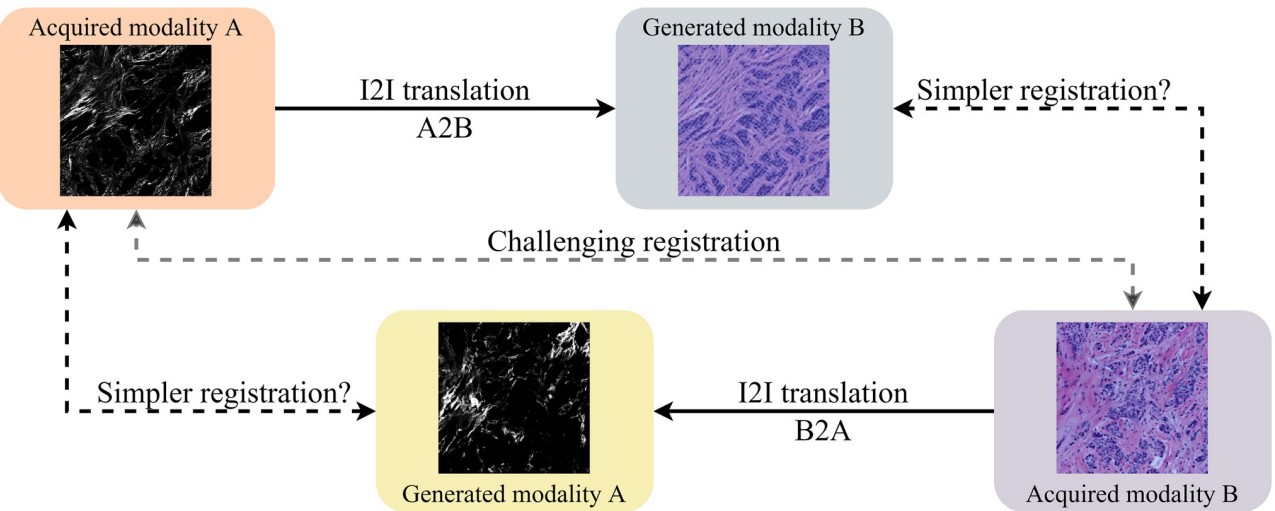

**Fig 1. Method overview.** Direct multimodal registration of distinctly different modalities is a challenging task (middle path). In this study, we evaluate if, instead, performing I2I translation of one modality to another may lead to a simpler monomodal registration problem (either of the two peripheral paths).

To fill this gap, and to facilitate knowledge transfer from the domain of multimodal natural image registration, we present an empirical study of the applicability of modern I2I translation methods for the task of multimodal medical and biomedical image registration. We compare the performance of four GAN-based methods, combined subsequently with two representative monomodal registration methods, to judge the effectiveness of modality translation on rigid registration of medical and biomedical images. We evaluate these method combinations on four publicly available multimodal datasets. To explore the effects of different types of data and their complexity, w.r.t. the task of registration, we include one relatively easy non-biomedical dataset, one Cytological dataset of medium difficulty, and one more challenging Histological dataset, all in 2D. Furthermore, we observe a 3D medical Radiological dataset containing T1- and T2-weighted MR images of brains. To establish reference performance, we include one classic intensity-based multimodal registration method, Mutual Information (MI) maximisation [24, 25], as well as two popular more recent multimodal approaches, Modality Independent Neighbourhood Descriptor (MIND) [26] and maximisation of similarity of Normalised Gradient Fields (NGF) [27]. On the Histological dataset, we also compare with a recent task-specific method [28] developed particularly for that type of data. In addition, we evaluate one recently proposed deep learning-based method for multimodal registration which uses a different approach to transform multimodal registration into a monomodal case [29]. We apply and evaluate this method on 3D (medical) data, which has not been done previously. We share our complete experimental setup as open-source, including method implementations, evaluation code, and all datasets to facilitate further reproducing and benchmarking. The framework can easily be extended by including additional methods, and can support further development and evaluation of approaches for registration of 2D and 3D medical and biomedical images.

## 2 Background and related work

Image registration methodologies can roughly be classified into intensity-based and feature-based methods, however with no strict border between the two; several modern approaches estimate and align more or less dense feature fields, aiming to capture good properties of both intensity- and feature-based methods. Learning-based approaches may, to different extents, be

utilised for directly solving the overall registration task (end-to-end), or for addressing different sub-tasks.

Intensity-based approaches typically formulate image registration as an optimisation problem, for which an image similarity measure (or inversely stated, a distance measure) is used to assess the quality of the alignment. Common similarity measures include negative Sum of Squared Differences (SSD) and Cross-Correlation (CC) for monomodal registration; Mutual Information (MI) [24] and Normalised Mutual Information [30] for both monomodal and multimodal registration; as well as measures which incorporate relevant additional information (e.g., gradient information or spatial displacement) extracted from the images [31, 32].

Several approaches for learning a suitable similarity measure from data have been proposed. A method to implicitly learn a multimodal similarity measure for the relatively easy multimodal scenario of registering T1 and T2-weighted Magnetic Resonance (MR) images is presented in [33], whereas [34] suggests training a Deep Convolutional Neural Network (DCNN) to predict correspondence of Computed Tomography and MR image patches and then use the classification score as a similarity measure.

Intensity-based approaches, which rely on iterative local optimisation of an image similarity measure, are often relatively slow [8], and the existence of many local optima in the search space reduces applicability when the displacement between images to be registered is large [32]. Several learning-based approaches have been proposed to speed up the optimisation task; representative examples include VoxelMorph [35], where an encoder-decoder network is used to directly predict a deformation field which maximises an image similarity measure, and Quicksilver [33], where a deep encoder-decoder network is used for patch-wise prediction of the momentum-parameterisation of a Large Deformation Diffeomorphic Metric Mapping model.

Feature-based methods detect salient structures in the images and find correspondence between them, to guide the registration. Popular methods for feature point detection and description include SIFT [36] and ORB [37]. Learning may be utilised to suitably adapt feature-based registration to data. Pioneering unsupervised approaches for learning features for image registration are presented in [38] (for MR data) and [39] (for natural scenes).

Compared to the iterative optimisation process of intensity-based methods, the feature-based approaches can be much faster and also more robust to variations in brightness and contrast, making this type of methods popular for computer vision and video tracking tasks. However, in multimodal setting, images typically convey complementary information; features that are present in one modality may be absent in another. Additionally, feature-based methods struggle with the general lack of salient structures in biomedical datasets. These shortcomings reduce the applicability of feature-based registration methods in multimodal biomedical scenarios.

Modality Independent Neighbourhood Descriptor (MIND) [26] combines feature- and intensity-based notions and extracts a dense set of multi-dimensional descriptors of each of the images of different modalities, utilising the concept of self-similarity. A standard similarity-based (monomodal) registration framework is then used with a standard monomodal similarity measure to establish correspondence between the generated representations.

Common representation of the images acquired by different modalities can also be learned; successful approaches include contrastive representation learning [29] and representation disentanglement [21]. Monomodal registration methods can then be applied to the resulting representations, as a suitable "common ground" between the modalities.

I2I translation (also known as image style transfer) refers to mapping images from a source domain to a target domain, while preserving important content properties of the source image. I2I translation can be used to translate images from one modality into images

appearing as if acquired in another modality. This enables to transform the multimodal registration problem into a, possibly easier, monomodal one, as illustrated in Fig 1. Prior to the tide of deep learning, I2I, also referred to as *image synthesis*, has been applied tentatively to multimodal registration problems [18, 40]. Despite the potential demonstrated in their promising performance, the full capability of this approach was still limited by the synthesising algorithms [19, 20]. I2I translation methods based on GANs [41] have, in recent years, shown impressive results [13–17].

I2I translation has found broad usage in biomedical areas [22], including virtual staining [42], stain style transfer [43], immunofluorescence marker inference [44], medical diagnosis [45, 46], and surgical training [47]. Usage for 3D medical images is reported for 3D brain tumour segmentation [48], and MRI modality generation [49] (which is conducted on the 2D slices of 3D volumes). In the recent CrossMoDA 2021 challenge, which involved domain adaptation of different MR modalities for tumour segmentation, methods based on I2I translation were the top performers [50]. A recent review [51], however, concludes that there is still a lack of 3D I2I translation methods and datasets suitable for their training. Even though studies utilising I2I translation in multimodal registration exist, proposing to either directly align the translated images [52–54] or indirectly incorporate the dissimilarity of modalities as a loss term in GANs [55, 56], these works are mostly application dependent due to the high variability of GANs' output. The full potential of I2I translation methods, and in particular their application in biomedical multimodal image registration, is yet to be explored, as also pointed out by recent overviews [57, 58].

Unlike registration of medical images, e.g., brain [21, 59], cardiac [60], or whole-body [61], where methods can rely on the outline of shapes (organs), biomedical images rarely convey shape outlines, making automatic registration much more difficult. One recent example where the biological structural content has been used to align microscopy images is presented in [28]; this highly specialised method for registration of Second Harmonic Generation (SHG) and Brightfield (BF) images relies on a priori knowledge of the sample properties and the segmented image content.

## 3 Considered methods

We select four state-of-the-art GAN-based I2I translation methods and one representation learning method designed specifically for the task of multimodal image registration. We subsequently combine these five methods with two monomodal registration approaches (one feature-based and one intensity-based). We compare the performance of these combinations with four selected reference registration methods, applicable directly to multimodal data.

### 3.1 Reference methods

To provide reference performance, we include three general multimodal image registration methods, as well as one special-purpose approach in our evaluation.

**3.1.1 Maximisation of Mutual Information.** The Mutual Information between images $A$ and $B$ is defined as

$$I(A, B) = \sum_{a \in \mathcal{A}} \sum_{b \in \mathcal{B}} p_{AB}(a, b) \log \left( \frac{p_{AB}(a, b)}{p_A(a) p_B(b)} \right) \tag{1}$$

where $\mathcal{A}$ and $\mathcal{B}$ denote the range of image $A$ and $B$ respectively, $p_A(a)$ denotes the marginal probability of value $a$ occurring in image $A$, and $p_{AB}(a, b)$ denotes the joint probability of values $a$ and $b$ co-occurring in overlapping points of $A$ and $B$. The probability mass functions $p_A$,

$p_B$, and $p_{AB}$ can be estimated empirically as the normalised frequencies of image intensities and pairwise image intensities.

Intuitively, MI measures the information that is (pixelwise) shared between the images, and is high when the images $A$ and $B$ are well aligned. Maximisation of MI is widely used and has shown good performance in a number of multimodal registration tasks.

**3.1.2 Maximisation of Similarity of Normalised Gradient Fields [27].** Registration based on Normalised Gradient Fields (NGF) relies on the assumption that parts of images are in correspondence (i.e., similar) if the orientation of changes (gradients) of the intensities at the corresponding locations in the two images is the same. Furthermore, in multimodal scenarios, the sign of the orientation is rarely of interest, since a single part of a specimen may have e.g. inverted intensities in two modalities. NGF-based registration is performed through maximisation of the squared dot-product of the normalised gradients of overlapping pixels. This optimisation problem can be solved by gradient-based optimisation, can be computed very efficiently, and has shown to perform well on medical images.

**3.1.3 MIND [26].** Modality Independent Neighbourhood Descriptors (MIND) are multi-spectral structural similarity representations based on self-similarity of small patches in a single image, computed in local neighbourhoods around the centre of each patch. The main assumption behind MIND is that the degree of self-similarity is similar across modalities, implying that the descriptors are similar as well, enabling the usage of optimisation methods suitable mainly for monomodal scenarios. The number of dimensions of the descriptors is equal to the number of displacements considered for each local patch; this number is selected by the user. The sizes of the local patches are determined by the standard deviation of the Gaussian kernel, $\sigma$, used to assign weights to the voxels of each patch.

**3.1.4 CurveAlign.** *CurveAlign* is the first automatic registration method designed specifically to register BF and SHG images of whole tissue micro-array (TMA) cores [28]. The registration is based on the segmentation of the collagen structures in BF images, which are then utilised to guide the alignment with the corresponding structures in the SHG images, by maximising MI between the modalities. We include this method in the evaluation performed on the considered Histological dataset (Sec. 4.3).

**3.1.5 VoxelMorph.** VoxelMorph is a recently proposed learning-based framework for deformable registration [35], which provides fast end-to-end transformation prediction. In our considered rigid multimodal registration scenario, its output deformation field is replaced by a transformation matrix and its smoothing term used to penalise local spatial variations is set to 0. MI is used as the loss function. However, VoxelMorph consistently under-performed in our rigid registration task (similar is also observed by [62]) and we decided to not include the related results, for clarity.

## 3.2 Modality translation methods

We introduce four I2I-translation methods and one representation learning approach, which are in focus of this study. The methods are selected considering their popularity in the community, generalisability to different applications, performance in general, diversity in terms of supervision, scalability to more modalities, and ease of usage.

**3.2.1 pix2pix [13].** pix2pix provides a general-purpose solution for I2I translation utilising conditional GANs (cGANs) [63], which use aligned image pairs during training. pix2pix consists of a generator $G$ and a discriminator $D$ which are trained in an adversarial manner: the generator $G$ is trained to generate fake images that cannot be distinguished from real ones by $D$, and $D$ is trained to best separate the fake and the real images. The objective

function is:

$$G^* = \arg \min_G \max_D \mathcal{L}_{cGAN}(G, D) + \lambda \mathcal{L}_{L1}(G), \qquad (2)$$

where the cGAN loss $\mathcal{L}_{cGAN}$ is to be minimised by $G$ and maximised by $D$ during training, and the pixel-wise regression loss $\mathcal{L}_{L1}$ aims to make the output images closer to the ground truth. Based on the assumption that the structures in one image should be roughly aligned between the input and output, the architecture of the generator $G$ is designed in a "U-Net" style [64].

pix2pix is included in our evaluation not only because of its pioneering role in the field, but also because it is regarded as a strong baseline framework [65, 66].

**3.2.2 CycleGAN [14].** Using cycle-consistency constraint has been shown to be an effective approach to overcome the need for aligned image pairs during training [67]. Cycle-consistent adversarial network (CycleGAN) is a well-known representative of this approach [68, 69]. Its modified version has demonstrated satisfying performance on the biomedical task of robust tissue segmentation [70] through stain transformation. Further modifications have also shown successes in other biomedical applications [46, 47]. The option of unsupervised training of CycleGANs makes them very attractive in biomedical applications.

The main idea behind CycleGANs is to enforce the image translation to be "cycle-consistent", i.e., if an image is translated from domain $\mathcal{X}$ to $\mathcal{Y}$ and then inversely translated from $\mathcal{Y}$ to $\mathcal{X}$, the output should be the same as the original image. To achieve this property, another generator $F : \mathcal{Y} \to \mathcal{X}$ is introduced to couple with $G$ such that $G$ and $F$ should be inverse mappings of each other. The used loss function combines two kinds of loss terms: the adversarial loss $\mathcal{L}_{GAN}$ to encourage the mapped domain to be close enough to the target [41], and the cycle-consistency loss $\mathcal{L}_{cyc}$ to enforce the invertibility of the mapping.

**3.2.3 DRIT++ [17].** Aiming for more diverse output from a single input, DRIT++ is a recently proposed GAN of higher complexity, that performs unsupervised I2I translation via disentangled representations. More specifically, DRIT++ learns to encode the input image into a domain-invariant content representation and a domain-specific attribute representation in latent spaces. The domain-invariant content representation captures the information that the two domains have in common, while the domain-specific attribute representation captures the typical features in each domain. The output images are then synthesised by combining the disentangled representations, where feature-wise transformation provides an option to enable larger shape variation.

For an I2I translation task across domains $\mathcal{X}$ and $\mathcal{Y}$, the framework of DRIT++ is comprised of a content encoder $E^c$, an attribute encoder $E^a$, a generator $G$ and a domain discriminator $D$ for each domain respectively, and a content discriminator $D^c$. To achieve the intended disentanglement of information, the weights between the last layer of $E^c_{\mathcal{X}}$ and $E^c_{\mathcal{Y}}$ and the first layer of $G_{\mathcal{X}}$ and $G_{\mathcal{Y}}$ are shared, following the assumption that the two images share some information in a common latent space [71]. The content encoders $\{E^c_{\mathcal{X}}, E^c_{\mathcal{Y}}\}$ are trained to make the encoded contents of the two domains indistinguishable from each other by the adversarial content discriminator $D^c$. DRIT++ also uses cross-cycle consistency loss, domain adversarial loss, self-reconstruction loss, and latent regression loss to encourage better quality of the generated images.

DRIT++ is included in this study because of its explicit extraction of shared information from the domains into a common latent space; notwithstanding its different purpose for output diversity, it is highly in accord with what is required for image registration and may therefore conceptually be likely to lead to good performance.

**3.2.4 StarGANv2 [15].** StarGANv2 is recently proposed to not only generate diverse output images, but also address the scalability to multiple domains. StarGANv2 comprises four modules: a style encoder $E_y$ to extract the style code $s = E_y(x_{ref})$ from a reference image $x_{ref}$, or alternatively, a mapping network $F_y$ to generate a style code $s = F_y(\mathbf{z})$ from an arbitrarily sampled vector $\mathbf{z}$ in the latent space; a generator $G$ to translate an input image $x$ and a domain-specific style $s$ into an output image $G(x, s)$; and a multi-task discriminator $D$ with multiple output branches $D_y$ to classify whether an output image is real or fake for each domain $y$. In addition to the diversity loss $\mathcal{L}_{ds}$ that enforces $G$ to discover meaningful style features from the image space, it also uses cycle-consistency loss $\mathcal{L}_{cyc}$ to preserve the content of the input image, style reconstruction loss $\mathcal{L}_{sty}$ and adversarial loss $\mathcal{L}_{adv}$ to ensure the output quality.

StarGANv2 is of specific interest due to the reported remarkably good visual quality of the generated images compared to the baseline models [17, 72], as well as the framework's increasing popularity in the community. Its ability to inject domain-specific style into a given input image could potentially simplify the registration problem when more than two modalities are considered.

**3.2.5 CoMIR [29].** CoMIR is a recently proposed representation learning method that produces (approximately) rotation equivariant image representations (typically of the same size as the input images), using an objective function based on noise-contrastive estimation (InfoNCE) [73].

For an arbitrary pair of images $(\boldsymbol{x}^1, \boldsymbol{x}^2)$ in the dataset $\mathcal{D}$, the InfoNCE loss is given by

$$\mathcal{L}^{opt}(\mathcal{D}) = -\mathbb{E}_{(\boldsymbol{x}^1, \boldsymbol{x}^2) \sim \mathcal{D}} \left[ \log \frac{\frac{p(\boldsymbol{x}^1, \boldsymbol{x}^2)}{p(\boldsymbol{x}^1)p(\boldsymbol{x}^2)}}{\frac{p(\boldsymbol{x}^1, \boldsymbol{x}^2)}{p(\boldsymbol{x}^1)p(\boldsymbol{x}^2)} + \sum_{\boldsymbol{x}_i \in \mathcal{D} \setminus \{\boldsymbol{x}\}} \frac{p(\boldsymbol{x}_i^1, \boldsymbol{x}_i^2)}{p(\boldsymbol{x}_i^1)p(\boldsymbol{x}_i^2)}} \right]. \tag{3}$$

The unknown ratio $\frac{p(\boldsymbol{x}^1, \boldsymbol{x}^2)}{p(\boldsymbol{x}^1)p(\boldsymbol{x}^2)}$ is approximated with the exponential $e^{h(y_i^1, y_i^2)/\tau}$, where $\boldsymbol{y}^1 = f_{\boldsymbol{\theta}_1}(\boldsymbol{x}^1)$ and $\boldsymbol{y}^2 = f_{\boldsymbol{\theta}_2}(\boldsymbol{x}^2)$ are the resp. modality translated images, and where $h(\boldsymbol{y}^1, \boldsymbol{y}^2)$ is a positive, symmetric 'critic' with a global maximum for $\boldsymbol{y}^1 = \boldsymbol{y}^2$ (e.g., negative mean squared difference (MSD), or cosine similarity) and $\tau$ is a scaling parameter. To equip the model with rotational equivariance, a loss term that implicitly enforces equivariance on $\mathcal{C}_4$ (the finite, cyclic, symmetry group of multiples of 90° rotations) is used:

$$h\left(T_1^{-1}(f_{\boldsymbol{\theta}_1}(T_1(\boldsymbol{x}_i^1))), T_2^{-1}(f_{\boldsymbol{\theta}_2}(T_2(\boldsymbol{x}_i^2)))\right), \tag{4}$$

where $T_1, T_2 \in \mathcal{C}_4$ are randomly sampled during training.

Given a batch of image pairs, the images of each modality are transformed using DCNNs $(f_{\boldsymbol{\theta}_1}$ and $f_{\boldsymbol{\theta}_2}$, one model per modality without weight-sharing) to a learned embedding space. The objective function serves to train the models to generate similar mappings for the paired images, while generating dissimilar mappings compared to all the other images in the batch. CoMIR requires pairs of aligned images for its training procedure, similarly to pix2pix. One advantage of the approach taken by CoMIR is that the requirement is not to learn to reproduce the appearance of either modality, but rather to learn a representation of the content (signal) present in both modalities, mapping corresponding image pairs into images with similar structure and appearance.

Combined with a monomodal registration framework which is applied to the learned representations, CoMIR has shown to exhibit good performance in the registration of (2D) multimodal images [29]. It is included in this study as a reference method based on a modality translation approach alternative to I2I, and its application is extended to 3D (medical) data.

## 3.3 Monomodal registration

We select two monomodal registration methods which have exhibited excellent performance in a range of applications: (i) SIFT—a feature-based approach which has for a long time dominated the field, demonstrating versatility and robustness in a large number of studies, and (ii) a recent method based on minimisation of a distance between images ($\alpha$-AMD), which has shown to constitute a top choice for medical and biomedical data. The selection of these generally applicable and well performing monomodal registration methods, to be combined with the modality translation approaches, enables to put focus particularly on the performance of the I2I translation methods in registration.

**3.3.1 SIFT-based registration.** Scale-invariant feature transform (SIFT) [36], is a 2D feature detector and descriptor which is invariant to image scaling, translation, rotation, and partially invariant to illumination changes and affine or 3D projection. It has found wide usage in a range of image matching and object recognition applications. Due to its popularity and generally good performance, SIFT is included in this study and evaluated on the 2D registration task (in combination with a suitable feature point matching and transformation estimation) on the original and generated images and their representations. The primary interest is in its performance in the (generated) monomodal scenarios.

**3.3.2 Registration based on $\alpha$-AMD.** A monomodal image registration method based on iterative gradient descent optimisation of $\alpha$-AMD has shown very good and robust performance [32, 74]. $\alpha$-AMD is a symmetric distance measure between images, which combines intensity and spatial information. Its properties enable to achieve a larger convergence region around the global optimum, compared to commonly used similarity measures that are based on statistics or intensity differences of overlapping points. Registration based on minimisation of $\alpha$-AMD has been successfully applied in a variety of monomodal scenarios [29, 32, 75], which motivates our decision to include it in this study for the registration of the generated monomodal image pairs.

# 4 Data

We use three publicly available 2D datasets and one 3D dataset in this study, to illustrate different applications (remote sensing, cytological, histological, and radiological imaging), combining different imaging modalities (Near-Infrared and RGB, Quantitative Phase Imaging and Fluorescence Microscopy, Second Harmonic Generation Microscopy and Bright Field Microscopy, and finally T1- and T2-weighted Magnetic Resonance Imaging) and introducing different levels of difficulty of image registration.

## 4.1 Zurich data

The Zurich dataset comprises 20 QuickBird-acquired images (with side lengths ranging from 622 to 1830 *px*) of the city of Zurich [76, 77]. Each image is composed of 4 channels, Near-Infrared (NIR), and three colour channels (R,G,B). Each channel is globally re-scaled to the range [0, 255]. The NIR channel is extracted as Modality A (Fig 2A), and the R-G-B channels are extracted jointly as Modality B (Fig 2B).

## 4.2 Cytological data

The Cytological dataset is composed of correlative time-lapse Quantitative Phase Images (QPI) and Fluorescence Images of prostatic cell lines acquired by the multimodal holographic microscope Q-PHASE (TESCAN, Brno, Czech Republic) [78, 79]. Three cell lines (DU-145, PNT1A, LNCaP) are exposed to cell death-inducing compounds (staurosporine, doxorubicin,

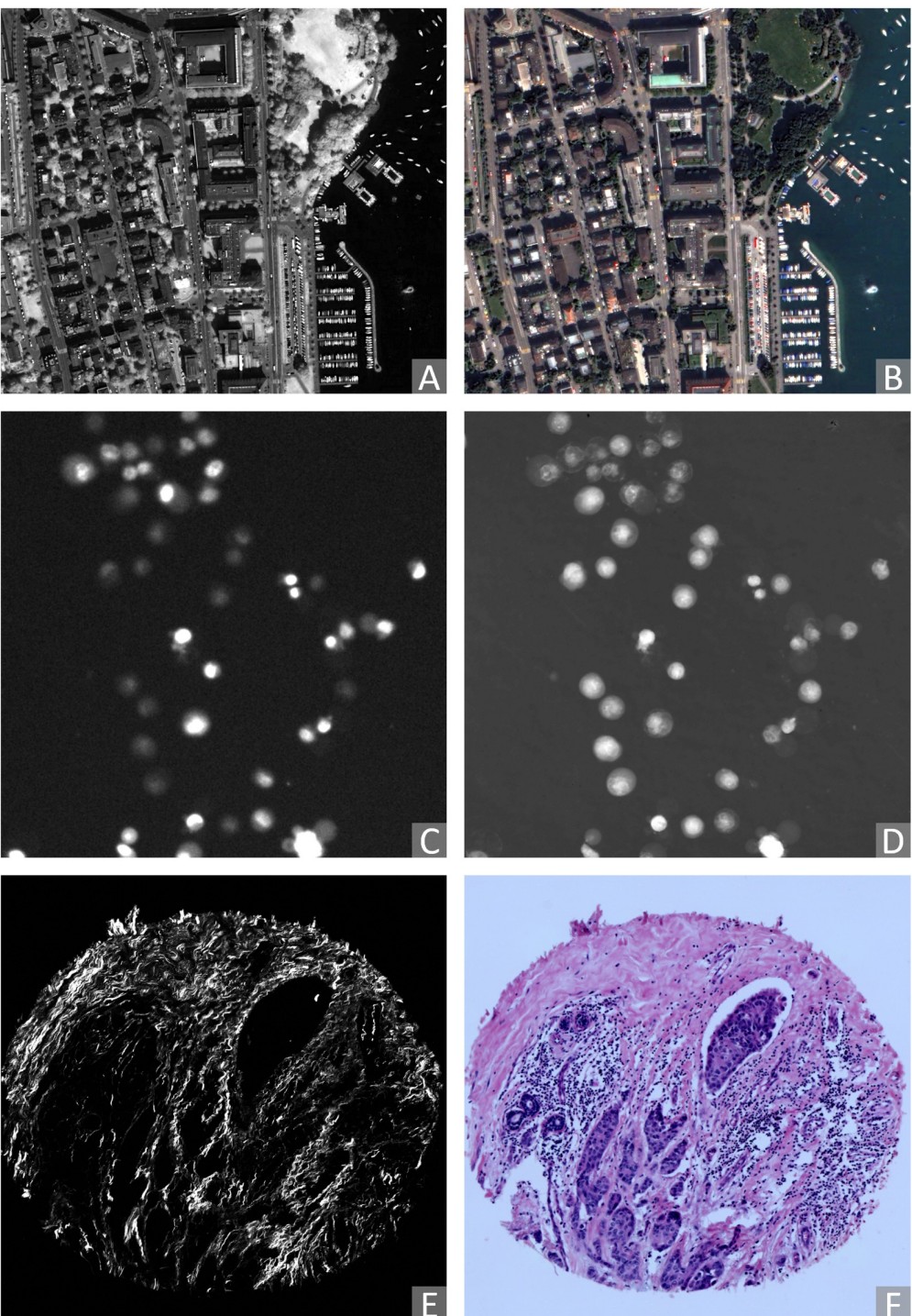

**Fig 2. Examples of pairs of original (2D) images acquired by different modalities considered in this study (contrast enhanced for visualisation).** Zurich (remote sensing) dataset: (A) modality A: NIR, (B) modality B: RGB. Cytological data: (C) modality A: Fluorescence microscopy, (D) modality B: QPI. Histological data: (E) modality A: SHG, (F) modality B: BF.

and black phosphorus) and captured at 7 different fields of view. The data consist of time-lapse stacks of several hundred $600 \times 600$ $px$ frames. Pixel values represent cell dry mass density in $pg/\mu^2$ in QPI images, and the amount of caspase-3/7 product accumulation in the fluorescence images (visualised using FITC 488 nm filter) [80]. For each time-lapse stack, the intensity values are globally re-scaled to the range [0, 255]. We visually inspected the image frames and confirmed that the signals of most cells are visible in the last 60 frames in each stack. We sparsely and evenly sample these frames such that the selected images are different enough from each other. Frames with indices 0, 15, 30, 45, 60 backwards from the last are extracted. We use the fluorescence images indicating the caspase-3/7 level as Modality A (Fig 2C) and corresponding QPI images as Modality B (Fig 2D).

### 4.3 Histological data

The Histological dataset comprises 206 aligned Second Harmonic Generation (SHG) and Bright-Field (BF) tissue micro-array (TMA) image pairs of size $2048 \times 2048$ $px$ [81]. The tissue sections are from patients with breast cancer, pancreatic cancer, and kidney cancer. BF images are acquired using an Aperio CS2 Digital Pathology Scanner (Leica Biosystems, Wetzlar, Germany) at 40× magnification. SHG images are acquired using a custom-built integrated SHG/BF imaging system described in [28]. Each SHG-BF image pair is registered by aligning manually marked landmark pairs. SHG and BF images are referred to as Modality A (Fig 2E) and B (Fig 2F) of the Histological dataset, respectively.

### 4.4 Radiological data

The Radiological dataset contains 3D magnetic resonance (MR) images of healthy brains, and is derived from the publicly available Retrospective Image Registration Evaluation (RIRE)-dataset [82]. The RIRE dataset includes CT, MR, and PET images of 18 patients in total [83]. We have focused on the provided pairs of T1- and T2-weighted MR images, due to non-available ground truth alignments of other combinations of modalities. After inspecting the alignments, we discarded 6 poorly aligned pairs. Therefore, here considered Radiological dataset includes 12 T1- and T2-weighted volume pairs, acquired using a Siemens SP 1.5 Tesla scanner. The volumes consist of 20–52 slices of $256 \times 256$ $px$ in $x$- and $y$-directions. Voxel sizes are 0.78–1.25 $mm$ in $x$ and $y$, and 3–4 $mm$ in $z$-directions. Further details about image acquisition and subsequent processing, performed to prepare the data for evaluation of registration, can be found in [82]. In our study, the intensity values are re-scaled within each volume to the range [0, 255]. The T1 modality is referred to as Modality A (Fig 3A), and the T2 modality is referred to as Modality B (Fig 3B).

### 4.5 Ground truth alignment of the image pairs

Alignment of multimodal image pairs is, in general, a very difficult task, for a human, as well as for an automated system, which makes it challenging to create ground truth data required for benchmarking, as well as for training the registration algorithms. Hybrid imaging—simultaneous acquisition of images of a sample (or a scene) by more than one modality—may provide such aligned pairs directly. Hybrid imaging is, however, available only to a limited extent. The Zurich, Cytological, and Radiological datasets are acquired by hybrid imaging, while aligned pairs of the Histological images are, as mentioned, generated semi-automatically.

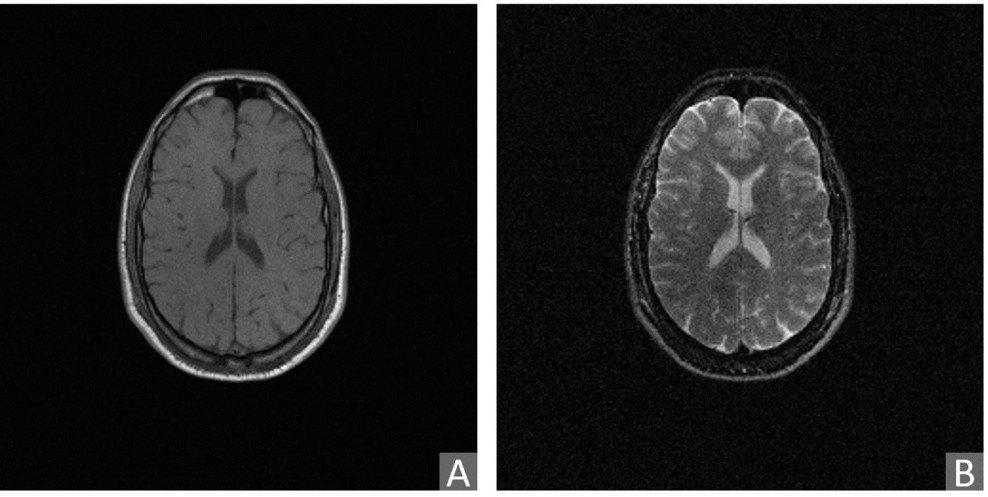

**Fig 3. Example of pair of (slices of) original 3D radiological images from the RIRE dataset acquired by different modalities considered in this study.** (A) Modality A: T1 MR image, (B) modality B: T2 MR image.

## 5 Experiments

### 5.1 Implementation of learning-based modality transfer methods

Aiming for a fair comparison, we use the same data augmentation for all the learning-based methods. For each 2D dataset, 5120 square non-rotated patches of size $362 \times 362$ *px* are randomly cropped from the training set. All models are trained with on-the-fly augmentation on top of the sampled patches. The augmentation includes, in a sequence: random horizontal flip ($p = 0.5$); random rotation ($p = 1$, $\theta \in [-180, 180˚)$) using either ($p = 0.33$) nearest neighbour, linear, or cubic interpolation; random Gaussian blur ($p = 0.5$) with a standard deviation $\sigma \in [0, 2.0]$; and centre-crop ($p = 1$) to size $256 \times 256$ *px* (thus, no padding required).

Apart from the shared augmentation, we make as few changes as possible compared to the default settings of the respective official released codes. After limited exploration of the parameter values close to the default ones, we have applied the following minor changes to the original implementations: (i) mini-batch sizes are (for reasons of processing speed) maximised under the limitation of 12 GB GPU memory, i.e., 8 for CoMIR, 64 for pix2pix and DRIT++, 4 for CycleGAN and StarGANv2; (ii) DRIT++ is set to use feature-wise transformation in the I2I translation of the Cytological, Histological and Radiological data, to enable the needed shape variation; this is not used on the Zurich dataset, where the shapes present in the images vary comparably little. (iii) For CoMIR, MSD is selected as the critic function. Its default scaling parameter $\tau = 0.5$ is used for all 2D datasets, while $\tau = 0.1$ is used for the 3D Radiological data.

Trainings are terminated after a certain number of iterations following the default settings of the models' official implementations, i.e., 200 epochs for pix2pix and CycleGAN, 1200 epochs for DRIT++, 100000 iterations for StarGANv2, 12800 iterations for CoMIR. All models are converged when training terminated.

During inference, to fit the network architectures of pix2pix, CycleGAN and StarGANv2, input image sizes are padded to minimum multiples of 256 pixels using "reflect" mode to reduce artefacts, e.g., the input image size $834 \times 834$ *px* of Histological data is padded to $1024 \times 1024$ *px*. The padded areas of the images resulting from I2I translation are cropped off before further processing. For DRIT++, to better fit the image registration context, a modified

version of the guided translation is used; instead of using the disentangled attribute representation of a random image from the other modality, the generator uses the disentangled attribute representation of the image that the current image is to be registered with. Similarly for Star-GANv2, the style code of the reference image guiding the translation is taken from the corresponding image in the target modality. Due to the restriction imposed by StarGANv2's architecture that the reference image can only be of size $256 \times 256\ px$, only the central areas of reference images are used to extract style code, to avoid resolution mismatch with training.

For the 3D Radiological data, all the modality transfer methods are trained with, and inference is performed on, axial slices along the z-axis. Each volume slice is resampled (B-spline interpolation) to physical space with 1 $mm^2$ square pixels (since the original volumes have different resolution). For training, each resampled slice is padded to size $362 \times 362\ px$ with the edge values. All models are trained with the same on-the-fly augmentation setting as for the 2D datasets, except for CoMIR where the final centre-crop ($p = 1$) was reduced to size $128 \times 128\ px$ so that the internal structures of the brains are better weighted against the skull outlines during training. After inference, the modality-transferred slices of the same volume are stacked back together to form a modality-transferred volume.

## 5.2 Implementation of image registration methods

Four of the six used registration methods—those based on MI, NGF, SIFT and $\alpha$-AMD—expect greyscale data. RGB images (Modality B of Zurich and the Histological dataset) are therefore converted to greyscale, using $L = 0.299R + 0.587G + 0.114$, before registration. The Euler angle rotation model (angles represented as radians) is used for all the registration methods. The parameters of each method are selected based on the values suggested in the original publications, suitably adjusted to account for differences in image sizes, general contrast levels, as well as level of detail of the image contents, relying on previous experience with the involved methods. For MI, NGF, MIND, and $\alpha$-AMD we employ a multi-start approach for the 2D registrations; three rotations, 0 and ±0.4 radians, are used as starting points, and the transformation reaching the lowest final distance is selected as the final registration output. For the 3D registrations, single-start is used throughout.

**5.2.1 Iterative registration by Mutual Information maximisation.** MI-based registration is performed using *SimpleElastix*—an industry-standard image registration library [84]. Adaptive stochastic gradient descent (ASGD) [85] is used as the optimiser and the maximum number of iterations is set to 1024. To avoid local minima, a multi-resolution strategy with a 4-level Gaussian scale space is used for Zurich, Cytological and Radiology data, while 6 levels are used for Histological data, considering the larger image size. In all experiments 32 histogram bins and bilinear/trilinear interpolation are used.

**5.2.2 Iterative registration by maximisation of similarity of Normalised Gradient Fields.** For optimisation of similarity of NGF we use the Autograd Image Registration Laboratory (AIRLab) framework [86], which supports a large number of objective functions (including similarity of NGF) and optimisation methods, multi-channel images, and multi-start optimisation. We use the ADAM optimizer, 4 pyramid levels with downsampling (8, 4, 2, 1) and smoothing with $\sigma = (15, 9, 5, 1)$, the iteration count was selected to be (2000, 1000, 500, 200). The step-size was chosen to be 0.01. Bilinear/trilinear interpolation is used.

**5.2.3 Iterative registration by MIND.** We evaluate optimisation of MIND both using MSD objective function (as is commonly done), and, in a novel combination of techniques, using the $\alpha$-AMD measure, where each channel is treated independently and the results are aggregated by summation. The standard deviation of the weighting Gaussian kernel is set to $\sigma = 0.5$. Local 4 neighborhoods are selected for 2D data, and local 6 neighborhoods are selected

for 3D data. For the MSD optimisation, the AIRLab framework is used with the same settings as for NGF concerning the number of pyramid levels, smoothing, iteration counts and starts. Bilinear/trilinear interpolation is used.

**5.2.4 CurveAlign.** The used implementation of *CurveAlign* follows the default settings in its V4.0 Beta version, except for restricting its default affine transformation to rigid.

**5.2.5 SIFT feature detection and matching.** SIFT feature detector is implemented using *OpenCV* [87]. The maximal number of retained feature points is limited to 500. Brute-force matching with cross-check is used to produce the best matches with the minimal number of outliers when there are enough matches. Other parameters remain at default settings. The rigid transformation between the two sets of matched coordinates is estimated using RANdom SAmple Consensus (RANSAC) algorithm [88] via the implementation in *scikit-image* [89]. A data point is considered an inlier if its residuals calculated for the estimated model is smaller than 2 *px*. For each image pair, the random sample selection procedure is iterated 100 times, and the final model is estimated using all inliers of the best model resulting from the performed iterations.

**5.2.6 $\alpha$-AMD.** $\alpha$-AMD is implemented based on the *Py-Alpha-AMD Registration Framework* (https://github.com/MIDA-group/py_alpha_amd_release). The following modifications are made compared to the settings in the example script: (i) Three resolution levels are used in a pyramid (coarse-to-fine) registration strategy for the 2D datasets, and four resolution levels for the 3D Radiological dataset, with sub-sampling factors set to (4, 2, 1), (for 2D data) and (8, 4, 2, 1) (for 3D data); Gaussian blur $\sigma$ is set to (12.0, 5.0, 1.0) for the 2D datasets and (5.0, 1.0, 1.0, 0.0) for the 3D Radiological data. The numbers of iterations per level are set to (900, 300, 60) for Zurich and Cytological data, to (3000, 1000, 200) for the larger Histological data, and (3000, 3000, 1000, 20) for the 3D Radiological data; (ii) Stochastic gradient descent (SGD) with momentum $\alpha = 0.9$ and a sampling fraction 0.01 is used. Step sizes for the first two resolution levels are set to 2 for the 2D datasets, while decreasing linearly from 2 to 0.2 for the last resolution level, and set to 1 for the first three resolution levels for the 3D dataset, while decreasing linearly from 1 to 0.1 for the last resolution level.

## 5.3 Evaluation

Our main objective is to evaluate the success of registration of multimodal images, utilising images translated from one modality to another as an intermediate result (to which monomodal registration is applied). We find it relevant to also evaluate the performance of the I2I translation methods on the observed datasets, and explore the effect of the quality of modality translation on the success of subsequent registration.

**5.3.1 Evaluation sets.** The Zurich dataset is divided into three sub-groups, enabling 3-folded cross-validation. The three groups are formed of the images with IDs: {7, 9, 20, 3, 15, 18}, {10, 1, 13, 4, 11, 6, 16}, {14, 8, 17, 5, 19, 12, 2}. Since the images vary in size, each image is subdivided into the maximal number of equal-sized non-overlapping regions such that each region can contain exactly one $300 \times 300$ *px* image patch. Then one $300 \times 300$ *px* image patch is extracted from the centre of each region. The particular 3-folded grouping followed by splitting leads to that each evaluation fold contains 72 test samples.

The Cytological data contain images from three different cell lines; we use all images from one cell line as one fold in 3-folded cross-validation. Each image in the dataset is subdivided from $600 \times 600$ *px* into $2 \times 2$ patches of size $300 \times 300$ *px*, so that there are 420 test samples in each evaluation fold.

For the Histological data, to avoid too easy registration relying on the circular border of the TMA cores, the evaluation images are created by cutting $834 \times 834$ *px* patches from the centres

of the original 134 TMA image pairs. This dataset has a defined split into training and test sets which we adhere to in our evaluation.

The evaluation set created from each of the three observed 2D image datasets consists of images undergone uniformly-distributed rigid transformations of increasing sizes of displacement. Each image patch is randomly rotated by an angle $\theta \in [-20, 20]$ degrees (with bi-linear interpolation), followed by translations in $x$ and $y$ directions by $t_x$ and $t_y$ pixels respectively, where $t_x$ and $t_y$ are randomly sampled within $[-28, 28]$ for Zurich and Cytological data, and within $[-80, 80]$ for the Histological data. To minimise border artefacts, the transformed patches are, for Zurich and Cytological data, padded using "reflect" mode, or, for Histological data, cropped at the appropriate position directly from the original larger images.

The 3D Radiological dataset is divided into three sub-groups, enabling 3-folded cross-validation. The groups are formed of the patients with IDs: {109, 106, 003, 006}, {108, 105, 007, 001}, {107, 102, 005, 009}. Since the Radiological dataset is non-isotropic (and also of varying resolution), we resample it using B-spline interpolation to 1 $mm^3$ cubic voxels before performing registration, taking explicit care to not resample twice; displaced volumes are transformed and resampled in one step. Reference sub-volumes of size $210 \times 210 \times 70$ voxels are cropped directly from centres of the (non-displaced) resampled volumes. Similarly as for the aforementioned 2D datasets, random (uniformly-distributed) transformations are composed of rotations $\theta_x$, $\theta_y \in [-4, 4]$ degrees around the x- and y-axes, rotation $\theta_z \in [-20, 20]$ degrees around the z-axis, translations $t_x$, $t_y \in [-19.6, 19.6]$ voxels in $x$ and $y$ directions and translation $t_z \in [-6.5, 6.5]$ voxels in $z$ direction. 40 rigid transformations of increasing sizes of displacement are applied to each volume. Transformed sub-volumes, of size $210 \times 210 \times 70$ voxels, are cropped from centres of the transformed and resampled volumes.

**5.3.2 Metrics.** *Modality translation.* Fréchet Inception Distance (FID) [90], a measure of similarity between two sets of images, has been widely used as an objective metric to assess the quality of GAN-generated images and has shown a high correlation with the subjective human judgement [15, 17, 58]. Lower FID (being a distance measure) indicates higher similarity of the images, corresponding to higher quality of the image translation. We use a PyTorch implementation of FID [91] to evaluate the quality of images generated by the considered modality translation methods.

*Registration.* We evaluate the performance of the considered registration approaches in terms of their success in recovering rigid transformations of varying size (amount of displacement) applied to one of the images in each of the aligned pairs. To quantify the success of registration, we define the spatial distance $D(I^1, I^2)$ between two image patches $I^1$ and $I^2$ as

$$D(I^1, I^2) = \frac{1}{n} \sum_{i=1}^{n} \left\| C_i^2 - C_i^1 \right\|_2, \tag{5}$$

where $C_i^1$ and $C_i^2$ respectively denote positions of the $n$ corner points of $I^1$ and $I^2$ ($n = 4$ for 2D datasets and $n = 8$ for the 3D dataset), when mapped into a common coordinate system.

The *initial displacement* $d_{\text{Init}}$ of a synthetic rigid transformation is the distance between a reference patch $I^{\text{Ref}}$ and its corresponding initially transformed patch $I^{\text{Init}}$: $d_{\text{Init}} = D(I^{\text{Ref}}, I^{\text{Init}})$. The *absolute registration error* $\epsilon$ is the (residual) distance between the reference patch $I^{\text{Ref}}$ and the transformed patch after registration $I^{Reg}$: $\epsilon = D(I^{\text{Ref}}, I^{Reg})$. The *relative registration error* $\delta$ is calculated as the percentage of absolute error to the width and height of the image patches: $\delta = (\epsilon/w) \times 100\%$, with $w = 300$ $px$ for Zurich and Cytological data, $w = 834$ $px$ for Histological data, and $w = \frac{1}{3}(210 + 210 + 70)$ $voxel$ for Radiological data. To summarise the competence of the registration of a sample, it is considered *successful* when the relative registration error $\delta < 2\%$. The *success rate* $\lambda$ is calculated as the ratio of succeed cases to total cases.

## 5.4 A framework for evaluation of multimodal registration methods

We share our complete experimental setup, including method implementations, evaluation code, and all datasets in order to facilitate further reproducing and benchmarking (https://github.com/MIDA-group/MultiRegEval).

Our created *Datasets for Evaluation of Multimodal Image Registration* is released on Zenodo with open access [92]. In total, it contains 864 image pairs created from the Zurich dataset, 5040 image pairs created from the Cytological dataset, 536 image pairs created from the Histological dataset, and metadata with scripts to create the 480 volume pairs from the Radiological dataset. Each image/volume pair consists of a reference patch $I^{\mathrm{Ref}}$ and its corresponding initially transformed patch $I^{\mathrm{Init}}$, in two modalities, along with the ground truth transformation parameters to recover the transformation. Scripts to compute the registration performance, to plot the overall results, and to generate more evaluation data with different setting are also included.

The framework can easily be extended by including additional methods, and can support further development and evaluation of approaches for registration of 2D and 3D medical and biomedical images. The provided datasets are large enough to enable training of deep learning based approaches, both those which require aligned image pairs and those which do not. Considering the observed lack of suitable benchmarks particularly designed for biomedical and medical applications and, at the same time, the increasing number of newly proposed registration methods, an option to perform comparative analysis under standardised conditions is highly beneficial and is expected to contribute to promotion of generally applicable and robust solutions.

# 6 Results

## 6.1 Performance of the modality translation methods

To summarise the modality translation performance, we present in Table 1, the computed FID values measured on the four datasets, for modality translations performed by each of the four I2I translation methods, in two directions (A to B, and B to A). The values in a row `method_A` indicate the distance between the distribution of images in the `method`-generated Modality A, and of those in the real Modality A.

We observe that the FID values reflect: (i) different levels of asymmetry of the I2I translation methods in treating translations in one, or the other directions (e.g., `cyc`, compared to `p2p` on Zurich dataset); (ii) rather varying stability of the results for the different methods, and the different datasets, as indicated by the large spread of standard deviations.

FID is also computed for CoMIR, indicating the difference between the distributions of the *representations* generated from Modality A and B. Although not directly comparable with the other values (computed on a rather different type of representation), these FID values, increasing from Zurich to the Histological dataset, are still informative in reflecting an increase in dataset difficulty.

The FID values in the row `B2A` indicate the difference between the original acquired multimodal images, for each of the four datasets. They can be seen as a reference value of the dissimilarity (FID) of the original images that, ultimately, are to be registered. We note that in some cases FID becomes higher after modality translation, compared to the initial `B2A` value.

Finally, the FID values computed within each of the modalities (between training and test splits), `A2A` and `B2B`, indicate the general scale of FID values for the respective modality. In most cases FID values for modalities A and B are rather similar (indicating that observed

**Table 1. Success of modality translation methods expressed in terms of Fréchet Inception Distance (FID).** Smaller is better. Standard deviations are taken over the 3 folds for Zurich, Cytological and Radiological data. `cyc`, `drit`, `p2p`, `star`, `comir` denote the methods CycleGAN, DRIT++, pix2pix, StarGANv2, and CoMIR, respectively. Suffix `_A` (resp. `_B`) denotes generated Modality A (resp. B). The best result achieved by an I2I translation method on each of the datasets is bolded. FID values between the initial considered multimodal image datasets (`B2A`), as well as between the training and testing splits within each modality (`A2A` and `B2B`) for each dataset, are included as references. FID values between the generated CoMIR representations are not directly comparable to those of the I2I translations, since the method generates a different (artificial) modality. Comparison with the values for the original multimodal data (`B2A`) confirms considerable reduction of FID.

| Dataset<br>Method | Zurich Data | Cytological Data | Histological Data | Radiological Data |
|---|---|---|---|---|
| cyc_A | 232.4±69.7 | **35.1±11.9** | 433.4 | 110.5±4.6 |
| cyc_B | **91.5±27.0** | 65.4±16.9 | 156.1 | **105.0±7.5** |
| drit_A | 182.9±3.3 | 63.1±17.4 | 125.3 | 244.8±16.0 |
| drit_B | 144.1±9.1 | 192.3±50.4 | 123.7 | 233.9±23.5 |
| p2p_A | 93.7±4.6 | 61.8±18.5 | **116.8** | 251.3±10.6 |
| p2p_B | 94.4±3.3 | 169.3±6.9 | 153.0 | 206.3±6.1 |
| star_A | 165.5±10.6 | 99.1±53.5 | 174.2 | 108.8±5.7 |
| star_B | 135.6±21.5 | 140.6±24.4 | 142.7 | 110.7±3.9 |
| comir | 17.7±8.0 | 61.5±11.1 | 91.0 | 89.9±20.7 |
| B2A | 155.3±15.4 | 145.0±17.6 | 341.0 | 134.4±8.7 |
| A2A | 113.7±0.9 | 48.8±27.1 | 99.1 | 76.9±2.2 |
| B2B | 104.3±3.7 | 82.8±20.4 | 102.6 | 86.4±4.0 |

asymmetries are due to the I2I translations) except for the Cytological dataset, where FID values for modality A are considerably lower than for modality B.

To complement these quantitative results, we show in Figs 4–7 a few examples of modality-translated images, by the evaluated methods, for each of the observed datasets. The examples illustrate some of the above-commented behaviours of the different methods.

Fig 4 shows examples from the Zurich dataset, with a clear correlation between the observed modalities. All the observed I2I translation methods reach reasonable to very good results. pix2pix generates the most realistic images, in both directions; it not only preserves the structural information but also maps the local intensities as desired. This is also reflected in the low FID values in Table 1. Other examples point out some issues: CycleGAN is successful in translating Modality A to B, but not the other way around. DRIT++ preserves structures, but is less successful regarding intensity mapping, whereas StarGANv2 fails to preserve geometry (straight lines) and generates a number of (colourful) noise pixels.

As can be seen in Figs 5–7, all the I2I translation methods exhibit more or less degraded performance when applied to the biomedical and medical datasets. Results on the Cytological dataset (Fig 5) indicate that the performance depends a lot on the "direction" (as indicated by FID as well); while all the methods give reasonably good output when translating from Modality B to A, the quality when translating Modality A to B tends to be much lower. This is particularly visible for pix2pix.

The Histological dataset (Fig 6), with distinctly different imaging modalities, elevates further issues and differences in the performance of the observed methods. Translated images exhibit non-realistic intensities and structures as well as "invented" details, particularly in translations from Modality A to B, where pix2pix completely fails. However, when instead translating from Modality B to A, pix2pix surprisingly captures the most structures, which is matched with the lowest FID value among the I2I translation methods on the Histological data.

On the 3D Radiological dataset (Fig 7), CycleGAN and StarGANv2 generate the most visually realistic output slices. This is also in accord with their low FID values in Table 1. pix2pix,

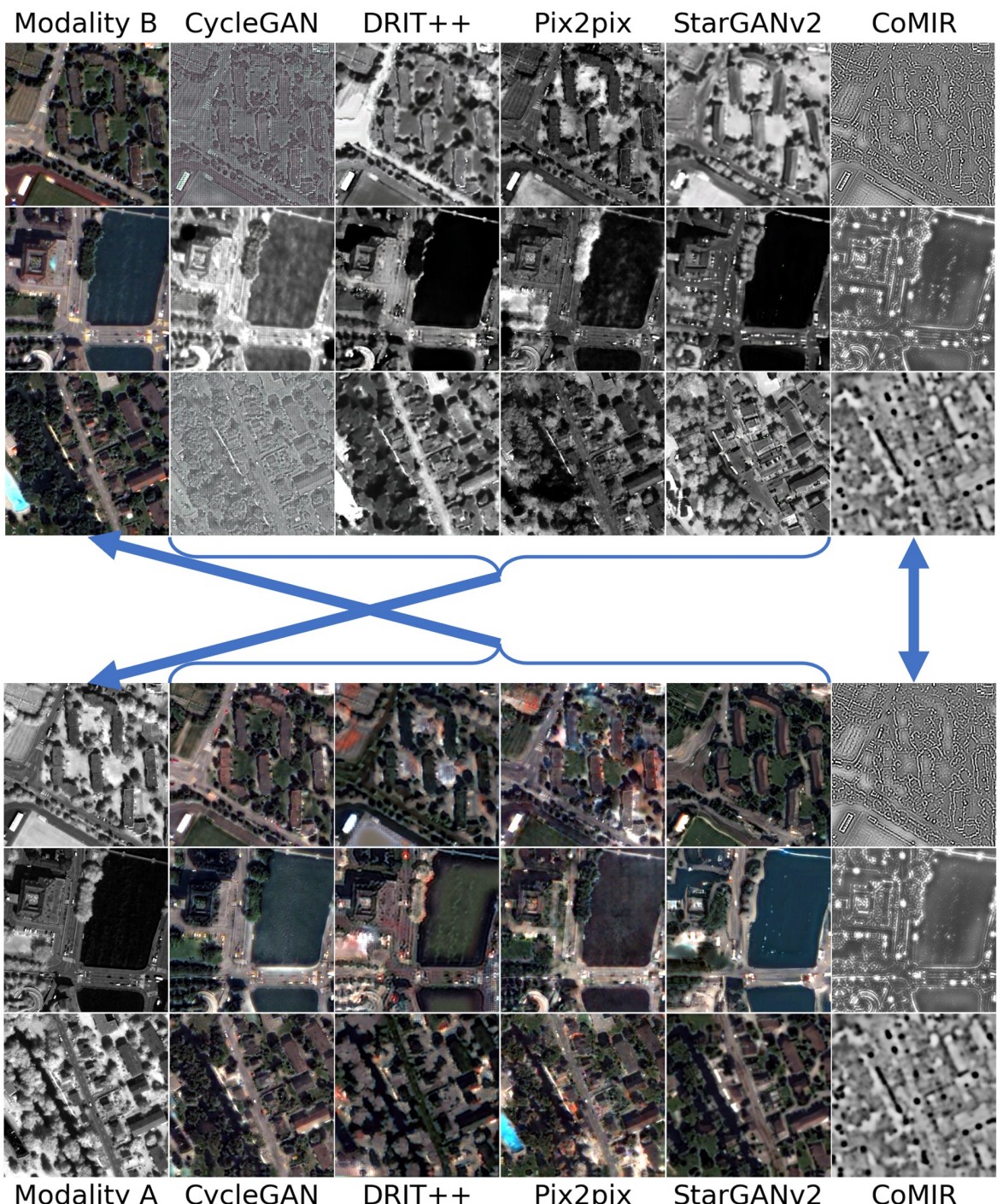

**Fig 4. Modality-translated image samples of the Zurich dataset by different evaluated methods (contrast-enhanced for visualisation).** Each row shows the results on one random image from each fold. Images in Columns 2–6 are generated from the images in (the corresponding row of) Column 1. Top block: Translations generated from Modality B. Bottom block: Translations generated from Modality A. The arrows indicate what to compare for visual inspection of the level of achieved similarity (pointing from generated images to the corresponding target of the learning process).

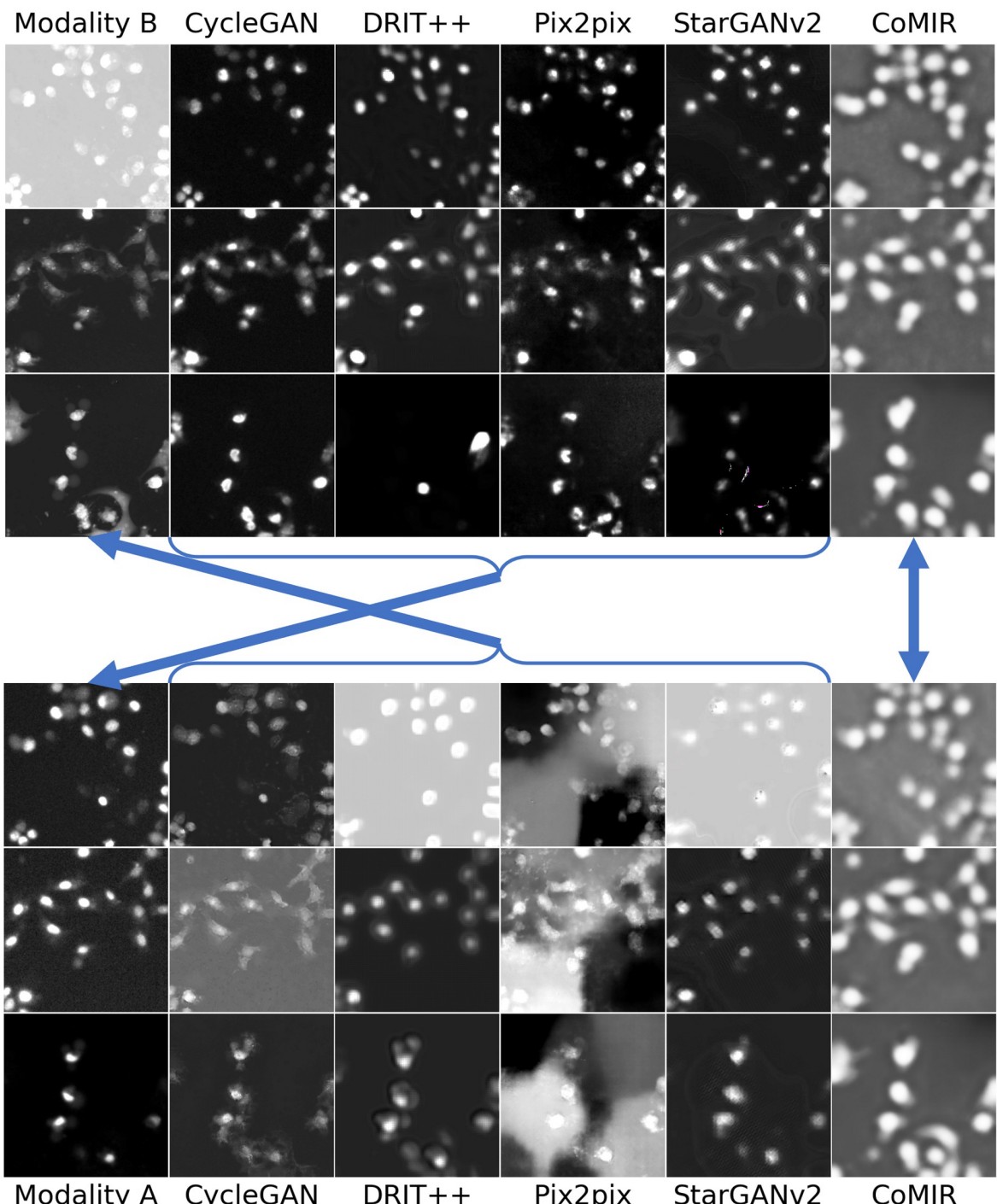

**Fig 5. Modality-translated image samples of the Cytological dataset by different evaluated methods (contrast-enhanced for visualisation).** Each row shows the results on one random image from each fold. Images in Columns 2–6 are generated from the images in (the corresponding row of) Column 1. Top block: Translations generated from Modality B. Bottom block: Translations generated from Modality A. The arrows indicate what to compare for visual inspection of the level of achieved similarity (pointing from generated images to the corresponding target of the learning process).

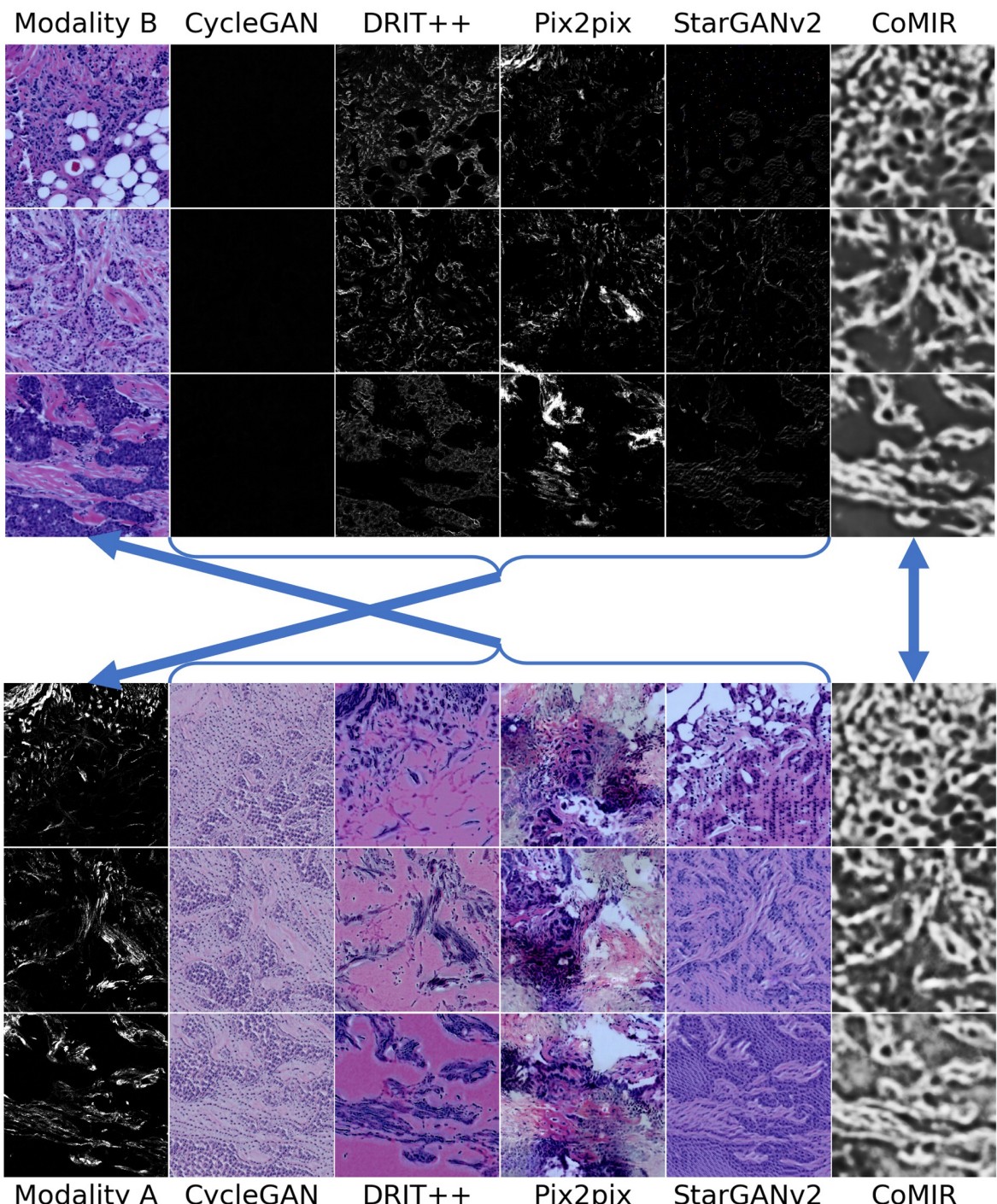

**Fig 6. Modality-translated image samples of the Histological dataset by different evaluated methods (contrast-enhanced for visualisation).** Each row shows the results on one random image from each fold. Images in Columns 2–6 are generated from the images in (the corresponding row of) Column 1. Top block: Translations generated from Modality B. Bottom block: Translations generated from Modality A. The arrows indicate what to compare for visual inspection of the level of achieved similarity (pointing from generated images to the corresponding target of the learning process).

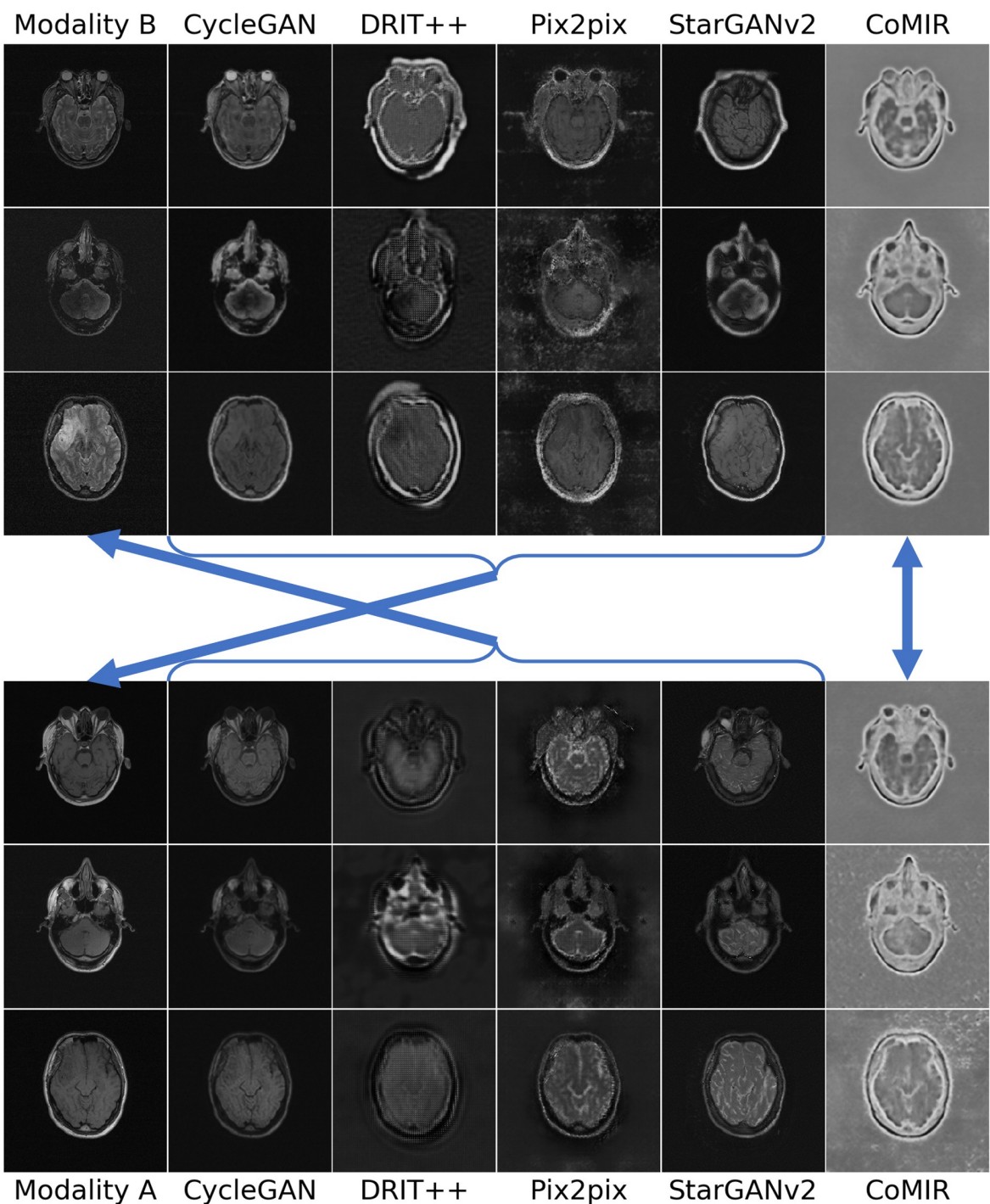

**Fig 7. Modality-translated image samples of the Radiological dataset by different evaluated methods.** Each row shows the results on one random slice from each fold. Images in Columns 2–6 are generated from the images in the corresponding row of Column 1. Top block: Translations generated from Modality B. Bottom block: Translations generated from Modality A. The arrows indicate what to compare for visual inspection of the level of achieved similarity (pointing from generated images to the corresponding target of the learning process).

although exhibiting unnatural artefacts and distortions, best captures the relative intensity of the eyeballs. CoMIR generates representations with a good balance of the extracted dominant structures (corresponding to the skull) and textural details (corresponding to soft tissue), while (similar to pix2pix) occasionally expressing artefacts in the surrounding background area.

We conclude that the visual assessment of the generated images most often supports the quantitative evaluation by FID, and clearly shows that none of the observed I2I translation methods exhibits stable performance, nor consistently outperforms the others. Even though we can not exclude the possibility that individually exploring the hyperparameter space and designing data augmentation schemes for each I2I translation method could lead to its slightly improved performance, we believe that such changes would not affect our main conclusion.

Performance of CoMIR, the non-I2I (but rather representation learning) modality translation method, is visually evaluated by assessing the similarity of the corresponding learned representations. We observe relatively high similarity for all four datasets, while slightly decreasing with increasing modality differentiation exhibited from Zurich to Histological dataset, as also observed in the FID values. Interestingly, despite relatively low intermodal diversity of the Radiological dataset (expressed by low B2A FID in Table 1), the different modality translation methods perform comparably poorly in the task of reducing it further; also here the best result is reached by CoMIR.

## 6.2 Registration performance

Plots showing registration success rate $\lambda$ for increasing initial displacement $d_{\mathrm{Init}}$ for the different evaluated combinations of methods are presented in Figs 8 and 9. The reference methods (MI, NGF, MIND, and CurveAlign) are included in both left and right columns of Fig 8 to facilitate easier comparison. We evaluated MIND both in combination with MSD and with $\alpha$-AMD minimisation and concluded that the novel combination of MIND and $\alpha$-AMD consistently performs better than the standard combination of MIND and MSD. For sake of visual clarity we therefore omit the latter from the graph. As already commented, we also exclude results obtained by VoxelMorph [35] due to its demonstrated consistently poor performance, which we attribute to the constraint of rigid registration (as also observed by [62]). In Table 2, we summarise the aggregated performance, over all the considered displacements. Here, we include the results obtained by MIND optimised both by MSD and $\alpha$-AMD.

A number of observations can be made from these results. Reference performance is established by iterative MI maximisation (black dashed line `MI`). We observe that, for smaller initial displacements, MI maximisation delivers outstanding performance on all four datasets. However, it is also apparent that the performance of MI (optimised by adaptive stochastic gradient descent) decreases fast as the initial displacement increases on the structure-rich Zurich and Histological datasets. This is not surprising, since MI is known to have a small region of attraction. On the other hand, its performance on the other two datasets is less impaired by the increasing initial displacement.

The yellow-green dotted curve `B2A` indicates the performance of monomodal registration approaches (SIFT and $\alpha$-AMD, in the respective subplots), applied directly to multimodal images. As expected, these methods underperform MI maximisation in all cases except for SIFT with relatively large transformations on the Zurich dataset.

MIND combined with $\alpha$-AMD minimisation shows better performance than MI on the structure-rich Zurich data, while falling behind on the other datasets. NGF performs comparably to MIND (combined with $\alpha$-AMD) on the Cytological data, while less favourably on the other datasets. MI consistently outperforms NGF.

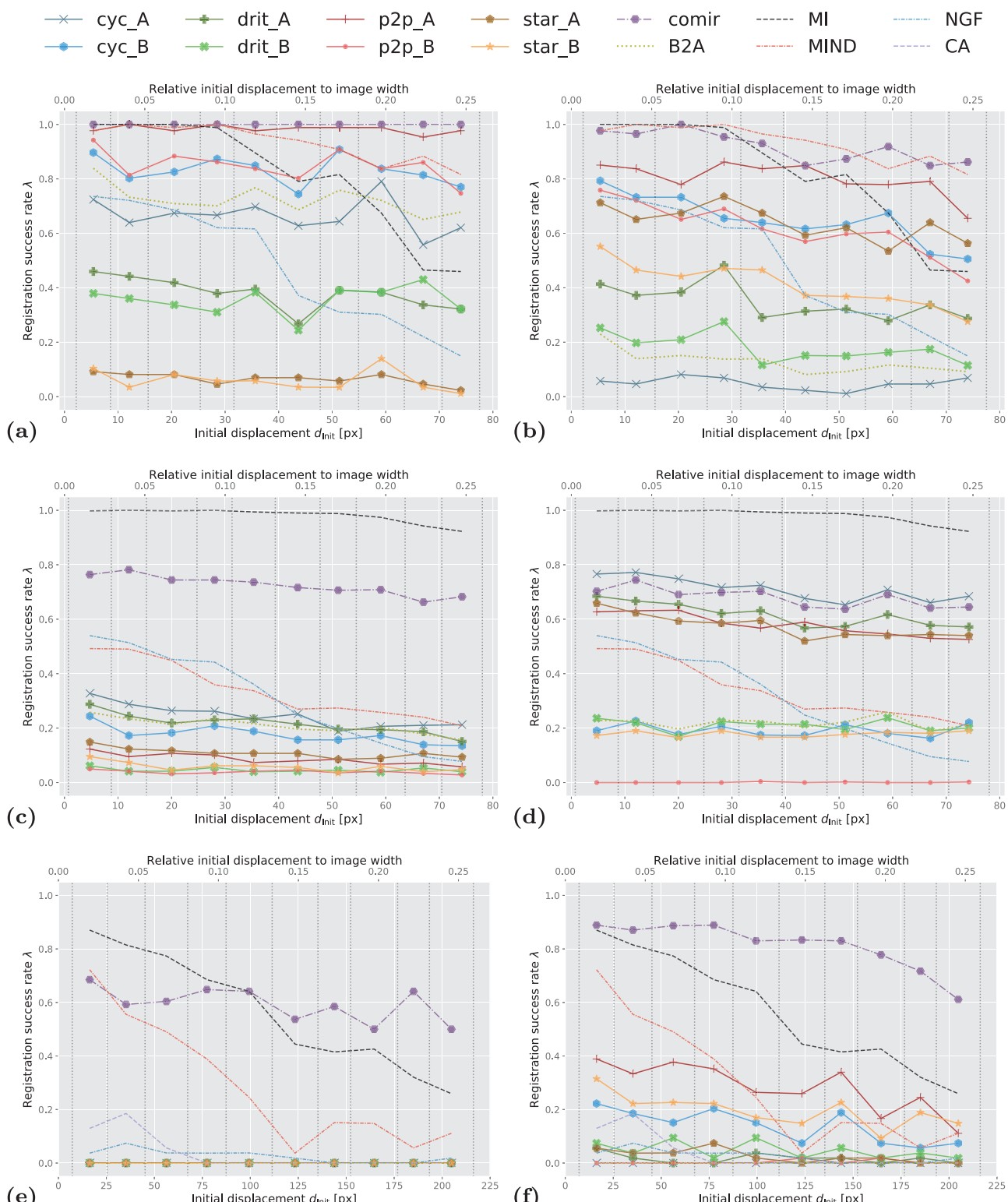

**Fig 8. Success rate of the observed registration approaches.** (A) SIFT on Zurich data, (B) $\alpha$-AMD on Zurich data, (C) SIFT on Cytological data, (D) $\alpha$-AMD on Cytological data, (E) SIFT on Histological data, (F) $\alpha$-AMD on Histological data. **x-axis**: initial displacement $d_{\text{Init}}$ between moving and fixed images, discretised into 10 equally sized bins (marked by vertical dotted lines). **y-axis**: success rate $\lambda$ within each bin (averaged over 3 folds for Zurich and Cytological data). In the legend, `cyc`, `drit`, `p2p`, `star` and `comir` denote CycleGAN, DRIT++, pix2pix, StarGANv2, and CoMIR methods respectively. Suffix `_A` (resp. `_B`) denotes that generated Modality A (resp. B) is used for the (monomodal) registration. `B2A` denotes registration of the

original multimodal images, without using any modality translation. `MI`, `MIND`, `NGF` and `CA` represent using MI maximisation, MIND, NGF and CurveAlign for registration, respectively.

The main focus of this study is to analyse the positioning of the remaining curves (with markers) presented in Figs 8 and 9, in comparison with the reference methods. First, we observe that, for each of the four datasets, there are a number of solid-line curves above the yellow-green dotted curve `B2A`, for all observed displacements. This demonstrates that some I2I translation methods indeed can help to approach a "monomodal case", and make the registration task easier when using the observed (monomodal) registration methods, SIFT and $\alpha$-AMD.

We also note that, similar as for the modality translation, the performances of the different registration methods show significant data-dependency. On the structure-rich Zurich data, SIFT performs rather well on modality translated images (and even on the original multimodal pairs). One exception is the StarGANv2 generated Zurich data, `star_A` and `star_B` in Fig 8A, where SIFT is completely failing. On the biomedical data, where features are less salient, $\alpha$-AMD, being an intensity-based method, tends to come out ahead. Fig 8E particularly indicates very poor performance of SIFT-based registration on the Histological data.

In general, the I2I translation methods followed by either SIFT (on 2D data) or $\alpha$-AMD (on 2D and 3D data) registration appear as far less sensitive to the initial displacement, compared to MI; in particular, several of them outperform MI on Zurich dataset for larger displacements.

What stands out in both Table 2 and in Figs 8 and 9, is the CoMIR (representation learning-based) modality translation method, which consistently outperforms the I2I translation

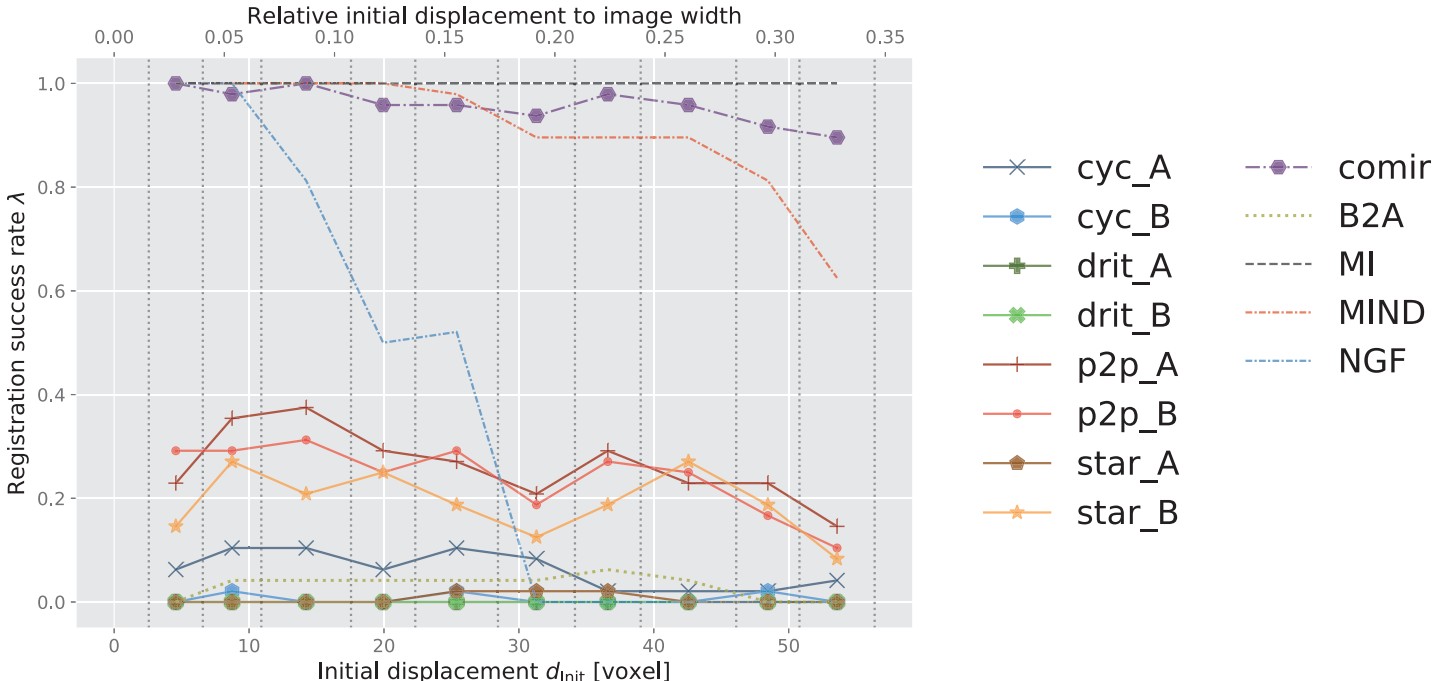

**Fig 9. Success rate of the observed 3D registration approaches on Radiological data.** *x*-axis: initial displacement $d_{\text{Init}}$ between moving and fixed images, discretised into 10 equally sized bins (marked by vertical dotted lines). *y*-axis: success rate λ within each bin (averaged over 3 folds). In the legend, `cyc`, `drit`, `p2p`, `star` and `comir` denote CycleGAN, DRIT++, pix2pix, StarGANv2, and CoMIR methods respectively. Suffix `_A` (resp. `_B`) denotes that generated Modality A (resp. B) is used for the (monomodal) registration. `B2A` denotes registration of the original multimodal images, without using any modality translation. `MI`, `MIND` and `NGF` represent using MI maximisation, MIND and maximisation of the similarity of NGF for registration, respectively.

**Table 2. Overall registration success rate (in percent) for the evaluated methods on four datasets.** Larger is better. The success rate λ is aggregated over all transformation levels for each dataset. Standard deviations are taken over the 3 folds for Zurich, Cytological and Radiological data. `cyc`, `drit`, `p2p`, `star`, `comir`, `MIND(α-AMD)`, `MIND(MSD)`, `NGF`, `MI` and `CA` denote the methods CycleGAN, DRIT++, pix2pix, StarGANv2, CoMIR, MIND+α-AMD-based registr., MIND+MSD-based registr., NGF, MI maximisation and CurveAlign, respectively. `_A` (resp. `_B`) denotes using generated Modality A (resp. B) for registration. `B2A` refers to the multimodal registration performance on the acquired images without modality translation. `MI`, `MIND` and `NGF` provide reference performance of good conventional multimodal registration methods. For each dataset, the best I2I-based approach, as well as the overall best performing (multimodal) approach, are bolded.

| Dataset | | Zurich Data | | Cytological Data | | Histological Data | | Radiological Data |
|---|---|---|---|---|---|---|---|---|
| Method | | α-AMD | SIFT | α-AMD | SIFT | α-AMD | SIFT | α-AMD |
| **Registration with modality translation** | | | | | | | | |
| I2I-based approaches | cyc_A | 4.9±2.1 | 66.4±18.8 | **71.1±5.8** | 24.4±6.2 | 0 | 0 | 6.2±3.9 |
| | bcyc_B | 65.0±8.4 | 83.2±3.1 | 19.2±2.8 | 17.6±2.5 | 13.8 | 0 | 0.6±0.0 |
| | drit_A | 34.8±5.4 | 38.0±7.9 | 61.6±16.2 | 21.6±3.6 | 1.7 | 0 | 0.0±0.0 |
| | drit_B | 18.1±3.1 | 35.4±3.5 | 21.0±9.0 | 4.6±1.3 | 4.7 | 0 | 0.0±0.0 |
| | p2p_A | 80.2±3.9 | **98.3±0.5** | 57.9±7.4 | 8.6±1.2 | **28.4** | 0 | **26.2±5.3** |
| | p2p_B | 61.5±4.7 | 85.0±5.0 | 0.1±0.1 | 3.8±2.0 | 0.4 | 0 | 24.2±5.7 |
| | star_A | 64.0±7.5 | 6.5±2.7 | 57.4±13.0 | 10.9±2.2 | 2.6 | 0 | 0.6±0.9 |
| | star_B | 41.1±3.6 | 5.9±0.5 | 17.8±4.9 | 5.8±0.6 | 19.6 | 0 | 19.2±12.5 |
| | comir | 91.8±7.7 | **100.0±0.0** | 68.0±14.0 | 72.5±7.1 | **81.3** | 59.3 | 95.8±5.0 |
| **Baseline without modality translation** | | | | | | | | |
| B2A | | 12.8±3.5 | 72.5±4.8 | 21.9±10.5 | 20.8±2.0 | 0 | 0 | 3.1±4.4 |
| **Reference direct multimodal registration methods** | | | | | | | | |
| MIND(α-AMD) | | 93.2±2.8 | | 33.8±5.2 | | 29.1 | | 91.0±3.6 |
| MIND(MSD) | | 41.4±1.6 | | 30.6±0.5 | | 9.5 | | 15.0±0.5 |
| NGF | | 47.3±7.4 | | 30.7±5.2 | | 2.6 | | 38.3±0.8 |
| MI | | 80.9±3.5 | | **98.1±0.8** | | 56.5 | | **100.0±0.0** |
| CA | | - | | - | | 3.7 | | - |

approaches. In combination with SIFT, only CoMIR manages on the Histological data (Fig 8E), while at the same time clearly outperforming the baseline for medium and large displacements. (Note that MI, MIND and NGF are not combined with SIFT, but are included in all the plots as reference methods.) On the Histological dataset, CoMIR combined with $\alpha$-AMD is the clear winner, while on the Cytological data MI is taking the top position, followed by CoMIR combined with either SIFT or $\alpha$-AMD, or CycleGAN combined with $\alpha$-AMD. On the Zurich dataset, CoMIR again performs best when combined with SIFT, and similarly well as MIND when combined with $\alpha$-AMD. On the 3D Radiological dataset, CoMIR (combined with $\alpha$-AMD) outperforms all I2I translation methods and demonstrates robustness to initial displacement, confirming to be a reliable approach to multimodal registration in 3D as well. While performing on par with MIND for smaller displacements, it better handles the task of recovering large transformations. However, MI maximisation is the top performing approach for this dataset.

Fig 8E and 8F also include the performance of the CurveAlign method, specially designed for registration of BF and SHG images in our Histological dataset. We observe that this method shows some limited success only for relatively small displacements, and falls behind MI, MIND and CoMIR, as well as several I2I translation methods when combined with $\alpha$-AMD.

## 6.3 Correlation between modality translation and registration

Intuitively, the more successful the modality translation is, the more accurate subsequent registration will be. Our results to a high extent confirm this: the observed performance of most

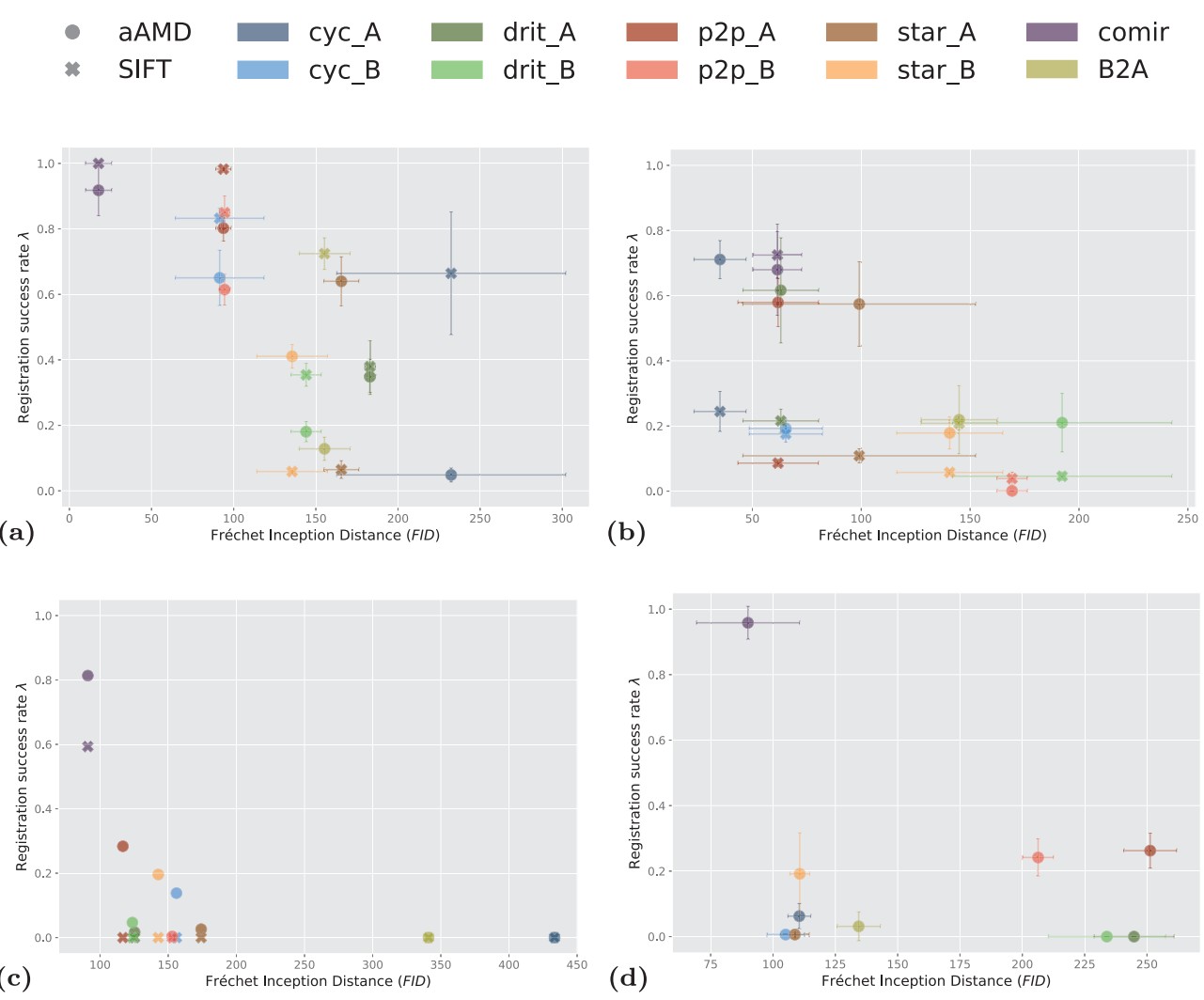

**Fig 10. Relation between average FID reached by a modality translation method and the success rate λ of the subsequent registration.** (A) On Zurich data, (B) on Cytological data, (C) on Histological data, (D) on Radiological data. In the legend, cyc, drit, p2p, star and comir denote the methods CycleGAN, DRIT++, pix2pix, StarGANv2 and CoMIR, respectively. Suffix _A (resp. _B) denotes that generated Modality A (resp. B) is used in (monomodal) registration. The marker style indicates whether $\alpha$-AMD (aAMD) or SIFT (SIFT) is used for the registration. The error-bars correspond to standard deviation computed over 3 folds for Zurich, Cytological and Radiological data.

combinations of methods is consistent with the appearance of the image samples in Figs 4–7. To visualise this consistency, we plot, in Fig 10, the relation between overall success rate λ, for each of the observed combinations of modality translation and registration, and the average quality of the used modality translation, as quantified by FID, for the four observed datasets. In Fig 10A, 10B and 10D, the error-bars show the standard deviations over the three folds of Zurich, Cytological and Radiological datasets.

The results show a clear common trend that lower FID (higher quality of the generated image) generally correlates with higher registration success rate. The large error-bars, in particular for cyc_A in Fig 10A and drit_B and star_A in Fig 10B, indicate that the corresponding GAN-based methods are highly unstable in modality translation. In spite of the relatively large error-bar of comir in Fig 10D, possibly caused by the artefacts visible in Fig 7, CoMIR still reaches the lowest average FID value and highest registration success rate (among

the modality translation based approaches) on the Radiological dataset. Also here we observe that the difference in performance between the two directions of modality translation may be rather high. Whereas the two modalities in the Cytological dataset visually differ less than those in Zurich and the Histological datasets, our results (summarised in Fig 10B) clearly show that all the four observed I2I translation methods have much more success in generating Modality A, than Modality B. This difference further propagates to registration, which is evidently more successful if based on generated Modality A, for both SIFT and $\alpha$-AMD.

## 6.4 Overview of the results

The presented results allow for the following summary:

- The observed I2I translation methods, followed by monomodal registration approaches, show to be applicable in less challenging multimodal registration scenarios, such as on the remote sensing (Zurich) dataset where the modalities display a relatively high degree of similarity in structure and appearance. However, the methods exhibit high instability and data dependence. The four observed I2I translation methods, with different properties and of different complexity, show varying performance on the observed datasets, including asymmetry w.r.t. to the direction of translation, without any of the methods standing out as a generally preferred choice.

- The unstable performance of I2I translation methods could be attributed to the images' geometry not being well preserved in the translated output. This problem is observed to be alleviated by utilising alignment information (such as pix2pix) or contrastive training (such as CoMIR).

- Among the observed modality translation methods, pix2pix and CoMIR require aligned image pairs during training. These methods also show superior performance, in particular on the highly structured Zurich dataset, compared to the remaining three.

- FID, as a measure of performance of I2I translation methods, shows to be a reasonably reliable predictor of the success of subsequent monomodal registration; I2I translation methods which manage to reach lower FID measured between the originally acquired and modality translated images can be expected to provide a better basis for a successful registration. I2I translation methods which are trained on non-aligned image pairs.

- The traditional registration approach based on MI maximisation exhibits very good performance, leaving behind all the observed I2I methods for sufficiently small displacements and excels on the Cytological and Radiological data, where there are more distinct objects to be registered. The combination of modality translation and monomodal registration, on the other hand, shows better stability w.r.t. size of displacement on the more structure rich Zurich and Histology datasets.

- Several I2I translation methods followed by $\alpha$-AMD outperform CurveAlign, a highly specialised method developed for registration of here observed Histological dataset.

- The novel combination of MIND and minimisation of $\alpha$-AMD as an objective function outperformed the original combination of MIND and minimisation of MSD in all observed scenarios.

- CoMIR, a representation-learning method developed for multimodal registration, here extended also to 3D data, exhibits overall best performance among the modality translation approaches, showing stability w.r.t. data, size of displacement, as well as the choice of subsequent monomodal registration method.

## 7 Conclusion

In this study, we investigate whether and to what extent I2I translation methods may facilitate multimodal biomedical image registration. We focus on 2D and 3D rigid transformations, finding the task challenging enough, while highly relevant, in particular for biomedical multimodal image data. We have selected four popular and widely used I2I translation methods with diverse properties and complexity. We believe that this selection gives a good insight into the potential of modality translation as a general approach to aid multimodal registration. Openly available multimodal biomedical datasets suitable for the evaluation of registration methods are very scarce. We use three 2D datasets (two of them published in connection with this study) and one 3D dataset, of varying complexity and different combinations of modalities, introducing a range of challenges relevant for performance evaluation.

From our experiments, we observe that I2I translation methods appear less successful in the context of multimodal registration than in some other relevant biomedical use-case scenarios (such as virtual staining, or image segmentation). However, a representation-learning approach (CoMIR), which maps the modalities to their established (learned) "common ground" instead of mapping one of them all the way to the other, shows to be a highly promising approach. We expect that its demonstrated successful debut on 3D medical data will inspire further development and applications. We also note that maximisation of Mutual Information still provides good multimodal registration performance in many situations under our experiment settings. However, other studies have shown that the effectiveness of such traditional multimodal registration methods degrades under real-world situations [21, 61].

This comparative study adds to the understanding of multimodal biomedical image registration methods from an empirical perspective. Last but not least, it establishes an open-source quantitative evaluation framework for multimodal biomedical registration based on publicly available datasets, which can be easily utilised for benchmarking, whereby we also hope to contribute to the openness and reproducibility of future scientific research.

## Supporting information

**S1 Appendix. List of abbreviations (and method names) used in the paper.**
(PDF)

## Acknowledgments

We thank Kevin Eliceiri (University of Wisconsin-Madison) and his team for kindly providing the data used to create the Histological dataset. We thank Jaromír Gumulec (Masaryk University) and his collaborators for kindly providing the Cytological dataset. We thank Michele Volpi and his collaborators for kindly providing the Zurich Summer Dataset. The RIRE dataset (images and the standard transformations were provided as part of the project "Retrospective Image Registration Evaluation", National Institutes of Health, Project Number 8R01EB002124-03, PI J. Michael Fitzpatrick, Vanderbilt University, Nashville, TN.

## Author Contributions

**Conceptualization:** Jiahao Lu, Johan Öfverstedt, Joakim Lindblad, Nataša Sladoje.

**Data curation:** Jiahao Lu, Johan Öfverstedt.

**Formal analysis:** Jiahao Lu, Johan Öfverstedt.

**Funding acquisition:** Joakim Lindblad, Nataša Sladoje.

**Investigation:** Jiahao Lu, Johan Öfverstedt, Joakim Lindblad, Nataša Sladoje.

**Methodology:** Jiahao Lu, Johan Öfverstedt, Joakim Lindblad, Nataša Sladoje.

**Project administration:** Joakim Lindblad, Nataša Sladoje.

**Resources:** Jiahao Lu, Johan Öfverstedt, Joakim Lindblad, Nataša Sladoje.

**Software:** Jiahao Lu, Johan Öfverstedt.

**Supervision:** Johan Öfverstedt, Joakim Lindblad, Nataša Sladoje.

**Validation:** Jiahao Lu, Johan Öfverstedt.

**Visualization:** Jiahao Lu.

**Writing – original draft:** Jiahao Lu, Johan Öfverstedt.

**Writing – review & editing:** Jiahao Lu, Johan Öfverstedt, Joakim Lindblad, Nataša Sladoje.

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
