## [Decision Letter · Decision Letter 0]

8 Jul 2022

PONE-D-22-07182Is image-to-image translation the panacea for multimodal image registration? A comparative studyPLOS ONE

Dear Dr. Lindblad,

Thank you for submitting your manuscript to PLOS ONE. After careful consideration, we feel that it has merit but does not fully meet PLOS ONE’s publication criteria as it currently stands. Therefore, we invite you to submit a revised version of the manuscript that addresses the points raised during the review process. In particular, both reviewers raised the need to cite additional references, as well as concerns about how the some of the evaluation experiments were conducted. 

We look forward to receiving your revised manuscript.

Kind regards,

Dzung Pham

Academic Editor

PLOS ONE

Journal Requirements:

The authors are financially supported by the Wallenberg AI, 906

Autonomous Systems and Software Program (WASP) AI-Math initiative, VINNOVA 907

(MedTech4Health project 2017-02447) and the Swedish Research Council (project 908

2017-04385).

However, funding information should not appear in the Acknowledgments section or other areas of your manuscript. We will only publish funding information present in the Funding Statement section of the online submission form. 

The authors are financially supported by the Wallenberg AI, Autonomous Systems and Software Program (WASP) AI-Math initiative, VINNOVA (MedTech4Health, project 2017-02447) and the Swedish Research Council (project 2017-04385). 

3. We note that Figures 2 and 4 in your submission contain satellite images which may be copyrighted. All PLOS content is published under the Creative Commons Attribution License (CC BY 4.0), which means that the manuscript, images, and Supporting Information files will be freely available online, and any third party is permitted to access, download, copy, distribute, and use these materials in any way, even commercially, with proper attribution. For these reasons, we cannot publish previously copyrighted maps or satellite images created using proprietary data, such as Google software (Google Maps, Street View, and Earth). For more information, see our copyright guidelines: http://journals.plos.org/plosone/s/licenses-and-copyright.

a. You may seek permission from the original copyright holder of Figures 2 and 4 to publish the content specifically under the CC BY 4.0 license.  

Reviewers' comments:

Reviewer's Responses to Questions

**Comments to the Author**

1. Is the manuscript technically sound, and do the data support the conclusions?

Reviewer #1: Yes

Reviewer #2: Yes

2. Has the statistical analysis been performed appropriately and rigorously? 

Reviewer #1: Yes

Reviewer #2: Yes

3. Have the authors made all data underlying the findings in their manuscript fully available?

Reviewer #1: Yes

Reviewer #2: Yes

4. Is the manuscript presented in an intelligible fashion and written in standard English?

Reviewer #1: Yes

Reviewer #2: Yes

5. Review Comments to the Author

Reviewer #1: The authors present an empirical study evaluating the efficacy of image translations (I2I) techniques for the purpose of multimodal registration. The study evaluates 4 recent deep learning based I2I techniques which are used with mono-modal registrations (SIFT+RANSAC and a-AMD) , and compared against four standard multi-modal methods as benchmarks. The methods were evaluated over 4 distinct biomedical imaging datasets, using Frechet Inception Distance (FID) to evaluate the image translation, and displacement accuracy to evaluation the registration.

Overall, I am impressed by this work. The authors were thorough in their analysis and explored a broad sample of techniques. The paper was also well written and reads very clearly. However, I have several concerns/questions I hope the authors can address:

1.) The introduction/background seems to be missing a discussion of image translation-based registration techniques that pre-dates the advent of deep learning in the field. Back then this was often referred to as image synthesis. Here are a few examples of these techniques, although many more were proposed in the literature:

Bogovic, J.A., Hanslovsky, P., Wong, A. and Saalfeld, S., 2016, April. Robust registration of calcium images by learned contrast synthesis. In 2016 IEEE 13th International Symposium on Biomedical Imaging (ISBI) (pp. 1123-1126). IEEE.

Chen, M., Carass, A., Jog, A., Lee, J., Roy, S. and Prince, J.L., 2017. Cross contrast multi-channel image registration using image synthesis for MR brain images. Medical image analysis, 36, pp.2-14.

Cao, X., Yang, J., Gao, Y., Wang, Q. and Shen, D., 2018. Region-adaptive deformable registration of CT/MRI pelvic images via learning-based image synthesis. IEEE Transactions on Image Processing, 27(7), pp.3500-3512.

2.) One topic that is rarely discussed in this area, which I hope the authors can touch upon, is the stochastic nature of deep learning based I2I techniques. That is, given the exact same train dataset, it is difficult to rebuild a model to provide the exact same image translation. While this problem isn’t unique to I2I models, it seems to have less of an impact on segmentation or classification networks where the model and results seem to converge quickly. In our experience, this issue gets amplified when further using the l2l model results as inputs for downstream registration, which seem to be very sensitive to this variance. Has the authors explored this limitation of these techniques? And how much impact do they think it has on their results? Is 3 folds sufficient to cover this variance?

3.) I am somewhat confused by the structuring of the comparisons reported in the experimental results. It seems the logical evaluation here would be to use each registration method with/without the image translation results as inputs, and swapping multimodal cost functions (MIND, MI, etc.) for monomodal ones (SSD, NCC, etc.) That way we can see the direct improvement that translating the modality has on the registration. However, this was only done for a-AMD and SIFT, and not fully for SIFT, since it wasn’t presented with a multimodal equivalent. The other methods were compared on just the raw images. This makes it difficult to tell if the loss/gains between the methods were due to the image translation, or differences between the registration technique themselves. For example, we know that Elastix is heavily tuned to perform well on MR images, that would explain the 100% registration accuracy on the MR dataset. However, that high performance is not necessarily because it used MI as a metric, there’s a good chance there are other heuristics built into the registration model that greatly help with the alignment. A more informative test would be to switch Elastix to using SSD instead of MI and use I2I results as inputs and see what happens. (And likewise for the other methods.)

4.) The authors mention that in Table 1, we can see some asymmetry depending on the translation direction. How do we know this difference is due to asymmetric performance of the image translation, and not a property of the different modalities? I.e. Is it possible the FID measures is just higher/lower in general for one modality over the other? Or is it normalized somehow to each modality?

5) In the conclusion, the authors make some strong statements about the efficacy of I2I based registration approaches relative to standard approaches (such as MI) based on their results. However, I think it is important to note that in this work, only intra-subject, rigid alignment, using simulated displacements were considered. This is a fairly limited registration task, and is not a great representation of the majority of real-world registration problems. Other studies have shown that once we start working with more complex transformations, cross subject/specimen data, or temporal shifts in the image, the effectiveness of traditional multi-modal measures such as MI starts falling dramatically, which is the driving motivation for many of the techniques cited by the authors.

Reviewer #2: This work tackles the well-known problem of multimodal registration, which lacks proper definitions for similarity functions between aligned images. Recently, due to the advent of many image-to-image translation methods, several methods propose to cast this multimodal registration problem as a monomodal problem, where several similarity functions have proved successful, by means of synthesising source domain images following the appearance of the target domain images.

The authors provide a nice trade-off between some of the standard metrics designed for multimodal registration and some methods using the intermediate I2I step with a monomodal similarity metric. Moreover, they add their recently proposed methods CoMIR, which in my opinion could be though also as an I2I but to a latent domain between the source and target domains.

The methods, datasets and experiments are well presented in the manuscript, even though sometimes it may be confusing due to the wealth of results and comparisons. I highlight the fact that the methods have been tested in 4 different medical imaging datasets.

However, I have some major comments and some minor comments that can be found in the attached PDF.

6. PLOS authors have the option to publish the peer review history of their article (what does this mean?). If published, this will include your full peer review and any attached files.

Reviewer #1: No

Reviewer #2: **Yes: **Adrià Casamitjana

---

## [Author Response · Author response to Decision Letter 0]

31 Aug 2022

Dear Dr Dzung Pham,

Thank you for giving serious consideration to our submission concerning image-to-image translation in the context of multimodal image registration. We are grateful to the associate editor and the two expert reviewers who have read and evaluated the merits of our manuscript. Following their recommendations, we have modified our study by: 

adding new reference results in Table 1;

adding discussions and citations of several related works;

modifying the presentation of our results.

Furthermore, we have thoroughly revised and updated the text, to further improve its informative content as well as its clarity, incorporating constructive and highly appreciated suggestions given by the reviewers. For clarity, all important changes are highlighted in yellow in the manuscript.

Please find enclosed our point-by-point responses to each of the comments raised by the reviewers and editor after the revision of our manuscript.

Sincerely,

Jiahao Lu, Johan Öfverstedt, Joakim Lindblad, Nataša Sladoje

To Journal Requirements

Authors’ response:

We have modified our style according to the provided guidance. Please let us know if we missed anything, and we will do our best to make further modifications to meet the required style.

“The authors are financially supported by the Wallenberg AI, Autonomous Systems and Software Program (WASP) AI-Math initiative, VINNOVA (MedTech4Health project 2017-02447) and the Swedish Research Council (project 2017-04385).”

However, funding information should not appear in the Acknowledgments section or other areas of your manuscript. We will only publish funding information present in the Funding Statement section of the online submission form. 

The authors are financially supported by the Wallenberg AI, Autonomous Systems and Software Program (WASP) AI-Math initiative, VINNOVA (MedTech4Health, project 2017-02447) and the Swedish Research Council (project 2017-04385). 

Authors’ response:

We have removed the funding information from our Acknowledgements Section. We are satisfied with the current Funding Statement and do not need to update it.

3. We note that Figures 2 and 4 in your submission contain satellite images which may be copyrighted. All PLOS content is published under the Creative Commons Attribution License (CC BY 4.0), which means that the manuscript, images, and Supporting Information files will be freely available online, and any third party is permitted to access, download, copy, distribute, and use these materials in any way, even commercially, with proper attribution. For these reasons, we cannot publish previously copyrighted maps or satellite images created using proprietary data, such as Google software (Google Maps, Street View, and Earth). For more information, see our copyright guidelines: http://journals.plos.org/plosone/s/licenses-and-copyright.

Authors’ response:

We have replaced the images included in Figures 2 and 4 with the new source published under CC BY 4.0 (https://doi.org/10.5281/zenodo.5914758), and updated our citation accordingly. 

To Reviewer #1

The authors present an empirical study evaluating the efficacy of image translations (I2I) techniques for the purpose of multimodal registration. The study evaluates 4 recent deep learning based I2I techniques which are used with mono-modal registrations (SIFT+RANSAC and a-AMD) , and compared against four standard multi-modal methods as benchmarks. The methods were evaluated over 4 distinct biomedical imaging datasets, using Frechet Inception Distance (FID) to evaluate the image translation, and displacement accuracy to evaluation the registration.

Overall, I am impressed by this work. The authors were thorough in their analysis and explored a broad sample of techniques. The paper was also well written and reads very clearly. However, I have several concerns/questions I hope the authors can address:

Authors’ response:

We thank the reviewer for recognising the contribution of our study and its relevance, as well as for the constructive comments and suggestions. We have no doubts that, by addressing them, we further increased the informative content and the overall quality of the manuscript. 

1.) The introduction/background seems to be missing a discussion of image translation-based registration techniques that pre-dates the advent of deep learning in the field. Back then this was often referred to as image synthesis. Here are a few examples of these techniques, although many more were proposed in the literature:

Bogovic, J.A., Hanslovsky, P., Wong, A. and Saalfeld, S., 2016, April. Robust registration of calcium images by learned contrast synthesis. In 2016 IEEE 13th International Symposium on Biomedical Imaging (ISBI) (pp. 1123-1126). IEEE.

Chen, M., Carass, A., Jog, A., Lee, J., Roy, S. and Prince, J.L., 2017. Cross contrast multi-channel image registration using image synthesis for MR brain images. Medical image analysis, 36, pp.2-14.

Cao, X., Yang, J., Gao, Y., Wang, Q. and Shen, D., 2018. Region-adaptive deformable registration of CT/MRI pelvic images via learning-based image synthesis. IEEE Transactions on Image Processing, 27(7), pp.3500-3512.

Authors’ response:

We find the reviewer’s suggestion very relevant. Therefore, we added discussions in both Section 1 (Introduction) and Section 2 (Background and related work), with all the mentioned works cited.

2.) One topic that is rarely discussed in this area, which I hope the authors can touch upon, is the stochastic nature of deep learning based I2I techniques. That is, given the exact same train dataset, it is difficult to rebuild a model to provide the exact same image translation. While this problem isn’t unique to I2I models, it seems to have less of an impact on segmentation or classification networks where the model and results seem to converge quickly. In our experience, this issue gets amplified when further using the l2l model results as inputs for downstream registration, which seem to be very sensitive to this variance. Has the authors explored this limitation of these techniques? And how much impact do they think it has on their results? Is 3 folds sufficient to cover this variance?

Authors’ response:

We appreciate the reviewer’s interest in this topic. We find it relevant, but somewhat out of the scope of our study; we see this as a part of the analysis and evaluation of each particular I2I method. To keep the focus of the study, we prefer to not further elaborate on the topic in the paper, but instead leave it to any interested reader to find further details about the used methods in the original publications, where the findings of their authors are reported. Our observations related to the reviewer’s question are summarised below: 

We agree with the reviewer that this topic is not widely discussed, but there are existing works touching upon this issue. In DRIT++[1] (one of the I2I-methods evaluated in our study), the authors’ reported variance values in their Table 1 are not so large, for any of their 4 evaluated metrics, as to expect a big impact on the downstream registration performance. Another cited work[2], which also uses I2I methods prior to deformable multimodal registration, reported variance values of 6 metrics on 2 datasets. It can be observed from their Tables 1 and 2 that the variance values of GAN-based I2I methods are close to, or smaller than, the ones of MIND and Elastix.

In our experiments, we also do not observe a large variance when training the I2I methods with exactly the same data. Do note that our variance values are not computed from multiple random runs given identical training data; the point of our 3-fold validation is to expose the variance when training with different data samples, which we find more relevant for real usage. What we observe and report on is, indeed as the reviewer mentioned, that the GAN-based I2I methods exhibit instability and strong data dependence. 

References:

[1] H.-Y. Lee et al., “DRIT++: Diverse Image-to-Image Translation via Disentangled Representations,” Int J Comput Vis, Feb. 2020.

[2] C. Qin et al., “Unsupervised Deformable Registration for Multi-modal Images via Disentangled Representations,” in Information Processing in Medical Imaging, 2019, pp. 249–261.

3.) I am somewhat confused by the structuring of the comparisons reported in the experimental results. It seems the logical evaluation here would be to use each registration method with/without the image translation results as inputs, and swapping multimodal cost functions (MIND, MI, etc.) for monomodal ones (SSD, NCC, etc.) That way we can see the direct improvement that translating the modality has on the registration. However, this was only done for a-AMD and SIFT, and not fully for SIFT, since it wasn’t presented with a multimodal equivalent. The other methods were compared on just the raw images. This makes it difficult to tell if the loss/gains between the methods were due to the image translation, or differences between the registration technique themselves. For example, we know that Elastix is heavily tuned to perform well on MR images, that would explain the 100% registration accuracy on the MR dataset. However, that high performance is not necessarily because it used MI as a metric, there’s a good chance there are other heuristics built into the registration model that greatly help with the alignment. A more informative test would be to switch Elastix to using SSD instead of MI and use I2I results as inputs and see what happens. (And likewise for the other methods.)

Authors’ response:

We thank the reviewer for pointing out these potential sources of ambiguity. Our evaluation strategy indeed does not exactly follow the structure envisioned by the reviewer, but in our opinion still offers a possibility to attribute changes in performance due to the I2I methods, separating them from the role of the particular used registration approaches. One important reason for not following the evaluation structure suggested by the reviewer is that a simple adjustment of the corresponding settings for monomodal and multimodal cases (such as changing the metric from MI in a multimodal case, to SSD in a monomodal case) is not possible for the majority of the used registration frameworks. The registration framework based on α-AMD is tailored for a particular distance measure (α-AMD), and does not support the direct use of other similarity/distance measures; it is shown in previous studies[1,2] that this particular distance measure has clear advantages over several alternatives (including both SSD and MI) in the monomodal registration context. Furthermore, MIND cannot be seen as a multimodal cost function (as formulated in the reviewer’s question), but the framework involves the generation of image representations (similarly to CoMIR, but without learning) prior to applying a selected metric on these representations to perform registration; in that sense, a discussion on the usage of multimodal and monomodal metrics, in the suggested way, does not appear appropriate. MIND can however be used in combination with different distance metrics, and we here evaluate both α-AMD and (the default) SSD/MSD for this task, where the prior clearly outperforms the latter. 

Based on the above, we have designed the evaluation to present the performance of a range of I2I-style methods in combination with (subsequently applied) different registration methods, as well as including several different types of baselines, both monomodal and multimodal; we believe that by doing so we managed to highlight, and support, the relevant and important results of the study. 

Let us also add that we agree with the reviewer that Elastix might be well optimised for a particular type of data (medical/brain images), but as can be observed from our results, the good performance of MI-based registration holds also for Cytological and partly for Histological data (where MI still clearly outperform all of the I2I-based paths, but at the same time stays far behind the best performing CoMIR combined with α-AMD). Therefore we feel confident in our conclusion that MI as such remains a good option for general multimodal registration. 

To respond to the reviewer’s question and the interesting initiated discussion, we below, as well as in the paper, try to further clarify our evaluation strategy: 

We set B2A as the baseline, which uses α-AMD and SIFT respectively to register the raw multimodal images, without intermediate image translations, or representations. (Neither of the two registration frameworks differentiates the settings for registration of multimodal vs monomodal data, and both are primarily designed with monomodal data in focus.) The direct effect of modality translation can be observed by comparing the I2I methods’ performance with this baseline. 

We include a comparison with registration frameworks based on MI maximisation, MIND (with two different similarity metrics) and NGF similarity maximization, in order to provide reference performance on each dataset of generally well-performing conventional multimodal registration methods, each in its own form which can be used out-of-the-box for end users. 

We have somewhat re-formulated our presentation of results in Table 2, aiming for a clearer comparison of the performance with/without modality translation. 

References:

[1] J. Öfverstedt, J. Lindblad, and N. Sladoje. Fast and Robust Symmetric Image Registration Based on Distances Combining Intensity and Spatial Information. IEEE Transactions on Image Processing, Vol. 27, No. 7, pp. 3584-3597, 2019.

[2] J. Öfverstedt, J. Lindblad, and N. Sladoje. INSPIRE: Intensity and Spatial Information-Based Deformable Image Registration. arXiv preprint: arXiv:2012.07208

4.) The authors mention that in Table 1, we can see some asymmetry depending on the translation direction. How do we know this difference is due to asymmetric performance of the image translation, and not a property of the different modalities? I.e. Is it possible the FID measures is just higher/lower in general for one modality over the other? Or is it normalized somehow to each modality?

Authors’ response:

We agree with the reviewer and find the concern that it is possible that the FID is just higher/lower in general for one modality over the other relevant and worth further investigation. Therefore we have evaluated the FID between the training and testing splits within each modality for each dataset to establish references, and we reported the obtained results in Table 1 (in the two added rows at the bottom). In most cases, the computed FIDs do not show any large differences for the different modalities within each of the datasets. An exception is the Cytological dataset where the FID is generally larger for modality B (82.8 ±20.4) than modality A (48.8 ±27.1), which is most likely attributed to partial intensity value overflows occurring during the data acquisition, as we observed during visual inspection. We removed our related (not totally well-founded) observation (ii) after Table 1.

5) In the conclusion, the authors make some strong statements about the efficacy of I2I based registration approaches relative to standard approaches (such as MI) based on their results. However, I think it is important to note that in this work, only intra-subject, rigid alignment, using simulated displacements were considered. This is a fairly limited registration task, and is not a great representation of the majority of real-world registration problems. Other studies have shown that once we start working with more complex transformations, cross subject/specimen data, or temporal shifts in the image, the effectiveness of traditional multi-modal measures such as MI starts falling dramatically, which is the driving motivation for many of the techniques cited by the authors.

Authors’ response:

We appreciate the reviewer’s opinion and find it very relevant. Hence we have modified our Conclusion section accordingly and mentioned the limitation of MI.

To Reviewer #2

This work tackles the well-known problem of multimodal registration, which lacks proper definitions for similarity functions between aligned images. Recently, due to the advent of many image-to-image translation methods, several methods propose to cast this multimodal registration problem as a monomodal problem, where several similarity functions have proved successful, by means of synthesising source domain images following the appearance of the target domain images.

The authors provide a nice trade-off between some of the standard metrics designed for multimodal registration and some methods using the intermediate I2I step with a monomodal similarity metric. Moreover, they add their recently proposed methods CoMIR, which in my opinion could be though also as an I2I but to a latent domain between the source and target domains.

The methods, datasets and experiments are well presented in the manuscript, even though sometimes it may be confusing due to the wealth of results and comparisons.

However, I have some major comments and some minor comments, both detailed below in a section-by-section basis. Before that, I wanted to highlight my main concern (repeated below). In my opinion, the experiments design may need a bit more of discussion or refactoring. I think that some readers may eventually say that the comparison is not entirely faire unless clarified by the authors. The I2I methods have been trained using different training data (not datasets, but different inputs, for example, aligned vs unaligned images), or hyperparameters (batch size, number of iterations). I acknowledge that all parameters cannot be set equally, but for a fair comparison one needs to decide which ones to keep and which not and justify this choice. In the case of this article, I think that the same training data (all methods sees the same aligned pairs in training), and batch size need to be kept constant for all methods. In contrast, the number of training iterations/epoch may be different based on the convergence of the methods (rather than a fixed number of epochs as it is now; in such case, I’d say it’s better to fix the same number of epochs for all methods).

Such decisions, independent of which ones, need to need properly justified. The authors mention all these experimental decisions but the reasons behind are not convincing to me.

Authors’ response:

We thank the reviewer for recognising the contribution of our study and its relevance, as well as for the constructive comments and suggestions. We have no doubts that, by addressing them, we further increased the informative content and the overall quality of the manuscript. We reply to the reviewer’s concerns in a point-to-point manner in the following paragraphs.

Background section:

I think that this paper (https://doi.org/10.1007/978-3-642-40811-3_79) asked a similar question about 10 years ago and may be mentioned in the introduction, as a very similar type of work. It can be also interesting to compare the conclusions from that article with the present in the conclusions.

Authors’ response:

We find the reviewer’s suggestion very relevant. Therefore, we added discussions in both Section 1 (Introduction) and Section 2 (Background and related work), with this paper and several other related works cited. The experiments in the mentioned article were conducted on already affinely-aligned images where only small local deformation exists, thus making a direct comparison of conclusion challenging. Their MRI T1-PD exemplar-based synthesis shows good visual quality, presumably due to rather correlated modalities. This indirectly supports our observation that the I2I-based approach exhibits rather high data dependence.

The distinction between intensity- and feature-based and hybrid registration methods seems appropriate for this work, since they are mainly focused on the multimodal scenario. They discuss several intensity-based and a few feature-based approaches. In the last paragraphs, they highlight several works that use I2I for registration of medical images. For completeness, I think that this part lacks:

In which category of the classification introduced by the authors do the I2I methods fall? (intensity, feature, hybrid).

Why do we need I2I? (justification)

What are the problems when training I2I translation methods? How is it addressed in the literature? (e.g., in 49 they use invariance between registration and translation)

Authors’ response:

We reply to the reviewer’s questions/suggestions as follows:

I2I methods do not actually fall into any category of registration methods, because I2I translation is used as a “pre-processing” step to convert the multimodal problem to a pseudo-monomodal one, and the classification into the mentioned categories should then be based on the subsequent registration approach. In our study, α-AMD and SIFT are used as representatives of intensity- and feature-based registration methods respectively, while both can be used in conjunction with the different I2I methods. 

Following the reviewer’s suggestion above, we extended the justification of introducing I2I translation to tackle the multimodal registration problem in Sec. 2, noting that with the recent advent of GAN-based I2I methods, the potential of this approach, which was previously limited by the synthesis methods, is yet to be explored. 

The main problem of using I2I methods in registration is caused by the unstable, data-dependent, and varying outputs of GANs; therefore the previous works (and suggested solutions) are mostly application dependent. We have added a statement on this. Since we are conducting a high-level comparative study of several GAN-based I2I methods, we prefer to avoid delving deeper into the analysis of more method- and application-specific details. An interested reader is referred to the cited original papers. 

Moreover, I think there are two concepts probably misused (happy to discuss with the authors so that they can convince me):

“Monomodal registration frameworks”. I don’t think a registration framework is monomodal or multimodal per se. It is the metric optimised that is more appropriate for monomodal or multimodal scenarios.

As an example, NiftyReg can use the SSD, LNCC or NMI metrics depending on the problem.

The authors define the MIND optimisation framework as a “similarity-based (monomodal) registration framework”, where they use the SSD between MIND representations of both moving and reference images, even though it can be used to align both monomodal and multimodal images.

In the same line, I’m not sure that “monomodal registration is easier”, there are just better defined losses (intensity-based, voxel-level) than in the multimodal case. For example, SSD of Local NCC.

Authors’ response:

We thank the reviewer for bringing up these relevant topics:

We agree with the reviewer’s point. To reduce confusion, we have modified our wording and phrasing accordingly to avoid statements such as “monomodal registration frameworks”. 

We consider the loss (similarity/distance used etc.) to be an integral part of the registration method, and thus the monomodal registration problem is typically easier because there (as also pointed out by the reviewer) exist better-defined losses. In addition, note that we only state that this is in general the case, since other factors also play a role in the overall difficulty.

Considered methods:

I think the second paragraph should be in the results section (less confusing for the reader). You actually mention it later as well.

“We also evaluated VoxelMorph [32] using MI as a loss function for the task of rigid multimodal registration. However, it consistently under-performed in our rigid registration task (similar is also observed by [57]) and we exclude the related results for clarity.”

Moreover, you could in few words say which VoxelMorph configuration did you test: SVF, deformation fields, rigid transform parameters (if exists)?

Authors’ response:

We appreciate and agree with the reviewer's comment. Thus we have removed this paragraph and added Sec. 3.1.5 to briefly introduce VoxelMorph and our performed modification.

Experiments:

Fair comparison of I2I methods: the same data augmentation is used (great!) but:

The data used is different! (depending on the need of aligned or not images).

The batch sizes are also different.

The number of iterations are also different and it is hard coded (it could be justified by convergence for this certain problem, but not regarding the default number of epochs, which are probably optimised for other tasks).

My thinking is that one could optimise each method and it may be right (e.g., run until convergence), but for a comparison to be fair some training parameters must be fixed and equal for all methods (e.g., the most restrictive method sets the parameter). The criteria should be clear and justified.

My special concern is with the use of aligned and not aligned training data, especially because training with aligned images seems more appropriate for the task at hand (which is registration, not I2I). Also, the method (at least for DRIT++ is made explicit but I think it’s also the case for CycleGAN, StarGANv2), is is trained with unpaired images but tested with paired images. And then in the results sections, CoMIR and Pix2Pix are the best performing ones.

The feature-wise transformation los DRIT++ should be briefly summarised in the methods section (3.2.3.) as you mention a slight modification and seems important for understanding the method comparison.

Authors’ response:

We appreciate the reviewer’s raised concerns, which indeed help us to improve the presentation of our study. We have made the following modifications to address each point above:

The considered I2I methods are selected based on their popularity in the community, and generally observed good performance in different applications, while ensuring diversity of the selection in terms of supervision, scalability to more modalities, and ease of usage. We have tried our best to make a fair comparison of such a diverse collection and adjusted the I2I methods where appropriate. Here we hope that some possible misunderstandings can be clarified:

The used data for all methods are actually exactly the same. The difference is how each method utilises the provided data. More specifically, “supervised” or “unsupervised” (in terms of utilisation of the aligned image pairs) is an intrinsic property of the I2I translation methods. Being unsupervised I2I translation methods, DRIT++, StarGANv2, and CycleGAN do not benefit from aligned pairs during training (these methods do not use that information even though they are presented with it). On the other hand, pix2pix and CoMIR need to utilise the alignment information for training and cannot be trained without. The qualitative advantage of the “unsupervised” I2I approaches is reasonable, while, as observed in our results, the quantitative benefits of the “supervised” ones are clearly shown. 

We can now understand why confusion regarding the used training data occurred and we have therefore removed the somewhat misleading sentence in Sec. 5.1; the way the training data is handled is not introduced by our implementation but intrinsically exists within the different methods. We clarify the supervised/unsupervised nature of the different I2I methods in Sec. 3.2 when introducing them.

Batch size has shown to be not a critical factor of performance[1]. We try to stay close to the default setting of the I2I methods while also optimising for our GPU resources. On the other hand, although keeping batch size constant makes sense when evaluating different configurations of a specific method, however, since different methods benefit differently from large batch sizes, using the same batch size for all methods is not necessarily fair. Our setting, limiting the GPU memory consumption for each method instead, reflects a more practical usage parameter and therefore may be more informative to the reader (who typically is, in a similar way, limited by the GPU memory).

We agree with the reviewer on this point. The convergence was indeed checked and confirmed for each training. We have modified our description in the manuscript. Yet we do not opt to set the parameters by the most restrictive method, but to give each method a fair chance to reach its best on this specific task.

We appreciate the reviewer’s suggestion and have modified our Sec. 3.2.3 accordingly.

References:

[1] S. Raschka, “No, We Don’t Have to Choose Batch Sizes As Powers Of 2,” Dr. Sebastian Raschka. [Online]. Available: https://sebastianraschka.com/blog/2022/batch-size-2.html. [Accessed: 21-Jul-2022].

Results:

Figure 8 needs indexing: A,B, etc…

Authors’ response:

We thank the reviewer for pointing this out. We find that using sub-captions provides improved clarity over indexing, however, this is not well handled in the PLOS review layout. We are keeping this technical issue in mind and assure to have it appropriately handled in the camera-ready version.

Conclusions:

One of the problems with I2I methods is that the topology of the generated image may not be preserved. Then, recent works such as (https://doi.org/10.1007/978-3-030-87592-3_5., citation 49 and even the authors’ method CoMIR could be thought as an being on this line) propose different options to ameliorate this fact. I think that, at least, this needs to be discussed somewhere in the paper (maybe linked to the previous comment about “problems with I2I translation methods”?). Maybe not in the conclusions but in sections 6.2 or 6.4.

Authors’ response:

We appreciate the reviewer’s suggestion and have added a paragraph of discussion in Sec. 6.4. We find that this mentioned recent work is indeed relevant and we have added a citation to it in our Sec. 2 when discussing related works using I2I for registration.

Minor comments:

1. In the introduction (lines 52-55) the authors seem to say that multimodal image registration using I2I and GAN-based approaches are mostly inexistent. I wouldn’t say so, and even the authors discuss many approaches in the following section.

Authors’ response:

We appreciate the reviewer’s suggestion and have modified our phrasing accordingly, to emphasise our work's uniqueness in quantitative evaluation and multiple datasets.

2. In Fig.1, I would not say that the registration from generated modality B to acquired modality B is simpler (same for modality A), but it’s monomodal. Since it is a monomodal registration, you could use standard pixel-based similarity metrics that are well defined in such scenarios. In the case of registering acquired modality A and B, the registration is not harder but multimodal; in this case there are not well-defined objective functions to optimise. The most widely used, as you say throughout the text, is the mutual information, but the performance is far from the monomodal scenarios. Moreover, what you depict here is more like a CycleGAN-based approach. I think it would be more clear to the reader just to show a forward transformation (or clarify it in the text and caption).

Authors’ response:

We appreciate the comment, but we disagree. Firstly, we do not claim that the monomodal approach is necessarily “simpler”. We use question marks to introduce the scientific question that our study is trying to answer, and have it clearly stated in the caption as possibly simpler. As aforementioned, we consider the loss an integral part of the registration problem; thus, monomodal is typically easier because better-defined losses exist. In addition, please note that there is no cycle in our figure. The image shows three optional paths of registration (as stated in the caption - "either of the two peripheral paths"). The modality translation can intrinsically be conducted in both directions, as we did in our experiments. Which direction to translate may depend on the information contained in the modalities, where asymmetry often exists. We think excluding one path would risk losing the information we intended to convey.

3. Good to have published the methods open-source in Github. I think it would be useful to have the link in the text (and not only in the abstract). Either in the introduction when you mention it, or in the Methods, section.

Authors’ response:

We appreciate the reviewer’s suggestion and have added a link in the first paragraph of Sec. 5.4, since we share not only the implementation of methods but also the whole experimental setup and evaluation protocol described there.

4. Please, uniform StarGANv2 and StarGAN-v2 naming convention.

Authors’ response:

We thank the reviewer for pointing this out. All namings have been unified to StarGANv2.

5. I think that MSD acronym is defined after the first use.

Authors’ response:

We thank the reviewer for pointing this out. The second (repeated) definition has been removed.

---

## [Decision Letter · Decision Letter 1]

2 Oct 2022

Is image-to-image translation the panacea for multimodal image registration? A comparative study

PONE-D-22-07182R1

Dear Dr. Lindblad,

We’re pleased to inform you that your manuscript has been judged scientifically suitable for publication and will be formally accepted for publication once it meets all outstanding technical requirements.

Kind regards,

Dzung Pham

Academic Editor

PLOS ONE

Additional Editor Comments (optional):

Reviewer 2 suggested that a sentence from Section 5.1 that was removed in the revision be put back. I leave it up to the authors whether they would like to add this sentence back in proofs.

Reviewers' comments:

Reviewer's Responses to Questions

**Comments to the Author**

1. If the authors have adequately addressed your comments raised in a previous round of review and you feel that this manuscript is now acceptable for publication, you may indicate that here to bypass the “Comments to the Author” section, enter your conflict of interest statement in the “Confidential to Editor” section, and submit your "Accept" recommendation.

Reviewer #1: All comments have been addressed

Reviewer #2: All comments have been addressed

2. Is the manuscript technically sound, and do the data support the conclusions?

Reviewer #1: Yes

Reviewer #2: Yes

3. Has the statistical analysis been performed appropriately and rigorously? 

Reviewer #1: Yes

Reviewer #2: Yes

4. Have the authors made all data underlying the findings in their manuscript fully available?

Reviewer #1: Yes

Reviewer #2: Yes

5. Is the manuscript presented in an intelligible fashion and written in standard English?

Reviewer #1: Yes

Reviewer #2: Yes

6. Review Comments to the Author

Reviewer #1: The authors have adequately addressed each of my major concerns regarding their work. I have no additional comments.

Reviewer #2: Dear authors,

Thanks for carefully addressing all concerns from both reviewers and the editors. I think the manuscript

has improved in readability and completeness. I mostly agree with all reviewer’s responses, and thanks for justifying your experimental decisions. I just want to add something related to the paired vs. unpaired discussion we began.

1.- Regarding your paragraph in the letter of response:

“Being unsupervised I2I translation methods, DRIT++, StarGANv2, and CycleGAN do not benefit from aligned pairs during training (these methods do not use that information even though they are presented with it). On the other hand, pix2pix and CoMIR need to utilise the alignment information for training and cannot be trained without. The qualitative advantage of the “unsupervised” I2I approaches is reasonable, while, as observed in our results, the quantitative benefits of the “supervised” ones are clearly shown.”

I completely agree that some methods (e.g., pix2pix and CoMIR) need paired training data by design and that others, such as CycleGAN, DRIT++ and StarGANv2 need not. However, the optimization of the latter seems easier when they are fed with paired images -- even though I don’t know if that is the case in practice (maybe it gets to a local minima that do not generalize well to unseen images).

In any case I would keep the sentence where you explain how you sample the images from both modalities depending on the method (Sec. 5.1). Apologies if my comment was confusing in the first place.

7. PLOS authors have the option to publish the peer review history of their article (what does this mean?). If published, this will include your full peer review and any attached files.

Reviewer #1: No

Reviewer #2: **Yes: **Adrià Casamitjana Díaz

---

## [Editor Report · Acceptance letter]

15 Nov 2022

PONE-D-22-07182R1 

Is image-to-image translation the panacea for multimodal image registration? A comparative study 

Dear Dr. Lindblad:

I'm pleased to inform you that your manuscript has been deemed suitable for publication in PLOS ONE. Congratulations! Your manuscript is now with our production department. 

Kind regards, 

on behalf of

Dr Dzung Pham 

Academic Editor

PLOS ONE